# CCVS: Context-aware Controllable Video Synthesis

**Guillaume Le Moing[1], [*]**     **Jean Ponce[1], [2]**     **Cordelia Schmid[1]**
[1]Inria and Department of Computer Science, ENS, CNRS, PSL Research University
[2]Center for Data Science, New York University

## Abstract

This presentation introduces a self-supervised learning approach to the synthesis of new video clips from old ones, with several new key elements for improved spatial resolution and realism: It conditions the synthesis process on contextual information for temporal continuity and ancillary information for fine control. The prediction model is doubly autoregressive, in the latent space of an autoencoder for forecasting, and in image space for updating contextual information, which is also used to enforce spatio-temporal consistency through a learnable optical flow module. Adversarial training of the autoencoder in the appearance and temporal domains is used to further improve the realism of its output. A quantizer inserted between the encoder and the transformer in charge of forecasting future frames in latent space (and its inverse inserted between the transformer and the decoder) adds even more flexibility by affording simple mechanisms for handling multimodal ancillary information for controlling the synthesis process (*e.g.*, a few sample frames, an audio track, a trajectory in image space) and taking into account the intrinsically uncertain nature of the future by allowing multiple predictions. Experiments with an implementation of the proposed approach give very good qualitative and quantitative results on multiple tasks and standard benchmarks.

## 1   Introduction

Feeding machines with extensive video content, and teaching them to create new samples on their own, may deepen their understanding of both the physical and social worlds. Video synthesis has numerous applications from content creation (*e.g.*, deblurring, slow motion) to human-robot interaction (*e.g.*, motion prediction). Despite the photo-realistic results of modern image synthesis models [38], video synthesis is still lagging behind due to the increased complexity of the additional temporal dimension.

An emerging trend is to use autoregressive models, for example transformer architectures [68], for their simplicity, and their ability to model long-range dependencies and learn from large volumes of data [4, 16]. First introduced for natural language processing (NLP) and then succesfully applied to visual data [18], the strength of transformers is grounded in a self-attention mechanism which considers all pairwise interactions within the data. The price to pay is a computational complexity which grows quadratically with the data size, which itself depends linearly on the temporal dimension and quadratically on the spatial resolution in the image domain. Although there have been some efforts to reduce the complexity of self-attention [11, 39, 50], using such methods directly on visual data is still limited to low resolutions and impractical without considerable computational power [9, 79].

Some recent works [20, 53] address this problem by using an autoencoder to compress the visual data, and apply the autoregressive model in the latent space. This greatly reduces the memory footprint and computational cost, yet, the greater the compression, the harder it is to faithfully reconstruct frames. The corresponding trade-offs may undermine the practical usability of these approaches. GANs [26] mitigate this issue by "hallucinating" plausible local details in image synthesis [20]. But latent video

---

[*]corresponding author: guillaume.le-moing@inria.fr

35th Conference on Neural Information Processing Systems (NeurIPS 2021).

transformers [53] decode frames independently, which prevents local details from being temporally coherent. Hence, using GANs in this setting may result in flickering effects in the synthesized videos.

We follow the problem decomposition from [20, 53], but introduce a more elaborate compression strategy with CCVS (for *Context-aware Controllable Video Synthesis*), an approach that takes advantage of "context" frames (*i.e.*, both input images and previously synthesized ones) to faithfully reconstruct new ones despite lossy compression. As shown in Figure 1, CCVS relies on optical flow estimation between context and new frames, within temporal skip connections, to let information be shared across timesteps. New content, which cannot be retrieved from context, is synthesized directly from latent compressed features, and adversarial training [26] is used to make up realistic details. Indeed, information propagates in CCVS to new frames as previously synthesized ones become part of the context. Like other video synthesis architectures based on autoregressive models [53, 79], CCVS can be used in many tasks besides future video prediction. Any data which can be expressed in the form of a fixed-size sequence of elements from a finite set (*aka*, tokens) can be processed by a transformer. This applies to video frames (here, via compression and quantization) and to other types of data. Hence, one can easily fuse modalities without having to build complex or task-specific architectures. This idea has been used to control image synthesis [20, 54], and we extend it to a variety of video synthesis tasks by guiding the prediction with different annotations as illustrated in Figure 2. Code, pretrained models, and video samples synthesized by our approach are available at the url `https://16lemoing.github.io/ccvs`. Our main contributions are as follows:

1. an optical flow mechanism within an autoencoder to better reconstruct frames from context,
2. the use of ancillary information to control latent temporal dynamics when synthesizing videos,
3. a performance on par with or better than the state of the art, while being more memory-efficient.

## 2 Related Work

**Video synthesis.** In its simplest form, videos are produced without prior information about their content. GAN-based approaches map Gaussian noise into a visually plausible succession of frames. For example, VGAN [71] and ProgressiveVGAN [1], adapt the traditional GAN framework [26] from image to video synthesis by simply using 3D instead of 2D convolutions. These approaches, including recent attempts such as $G^3AN$ [76], are computationally expensive, and, by nature, restricted to synthesizing a fixed number of frames due to the constraints of their architecture. Other approaches predict latent motion vectors with a CNN [56], or a recurrent neural network (RNN) [12, 57, 65], and generate frames with individual 2D operations. To avoid the shortcomings of information loss in the sequential processing of RNNs, we use an attention-based autoregressive model instead. Since we forecast temporal dynamics in a compressed space, a large temporal window can be used when predicting new frames, without having to resort to expensive 3D computations. Previous works have attempted to scale video synthesis to higher resolution by using progressive training [1], subsampling [56, 57], reducing the dimension of the discriminator [36], or redefining the task as finding a trajectory in the latent space of a pre-trained image generator [63]. Compression, together with efficient context-aware reconstruction allows us to synthesize videos at high resolution.

**Controllable video synthesis.** Some of the aforementioned works [12, 56, 57, 76] handle synthesis conditioned on a class label. Another popular control is to use a few priming frames to set off the generation process. This task, known as *future video prediction*, has received a lot of attention recently. Methods based on variational autoencoders (VAE) [2, 15] have been proposed to account for the stochastic nature of future forecasting, *i.e.*, the plurality of possible continuations, disregarded in deterministic predictive models [21, 47, 70]. Yet, their blurry predictions have motivated the incorporation of adversarial training [44], hierarchical architectures [7, 80], fully latent temporal models [22], or normalizing flows [42]. Another line of work infers spatial transformation parameters (*e.g.*, optical flow), and predicts the future by warping past frames (*i.e.*, grid sampling and interpolation as in [33]) in RGB space [21, 24, 28, 72, 81] or in feature space [45], typically using a refinement step to handle occlusions. Lately, autoregressive methods [53, 79] leveraging a self-attention mechanism [68] have also been applied to this task. Our method benefits from the context-efficiency of approaches based on spatial transformation modules and the modeling power of autoregressive networks. In the meantime, other forms of control with interesting applications have emerged. Point-to-point generation [75], a variant of future video prediction, specifies both start and end frames. State or action-conditioned synthesis [28, 48] guides the frame-by-frame evolution with high-level commands.

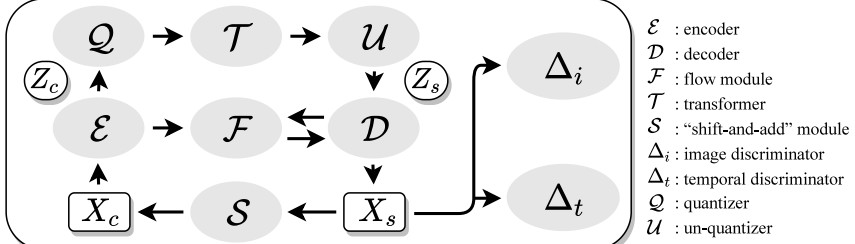

Figure 1: Proposed architecture. Here $X_c$ and $X_s$ respectively stand for the (input) context and (output) synthesized video. Learnable encoder and decoder modules $\mathcal{E}$ and $\mathcal{D}$ are linked by a learnable flow estimation module $\mathcal{F}$ ensuring spatio-temporal consistency between context and synthesized frames. The architecture is doubly autoregressive, with a transformer $\mathcal{T}$ responsible for predicting the features $Z_s$ associated with future frames $X_s$ from the features $Z_c$ associated with context frames $X_c$, and a simple, parameterless, "shift-and-add" module $\mathcal{S}$ updating $X_c$ as each new frame is generated. The architecture is trained in two steps: The parameters of $\mathcal{E}$, $\mathcal{D}$ and $\mathcal{F}$ are first estimated (without any future forecasting from the transformer) in an adversarial manner using two discriminators $\Delta_i$ and $\Delta_t$ to ensure that the frames synthesized are both realistic ($\Delta_i$) and temporally coherent ($\Delta_t$). The two discriminators are then discarded, and the parameters of the transformer $\mathcal{T}$ are estimated with $\mathcal{E}$, $\mathcal{F}$ and $\mathcal{D}$ frozen. At inference time, the transformer is used only once, latter frames being estimated in an autoregressive manner by the quadruple $\mathcal{D}\,\mathcal{S}$, $\mathcal{E}$, $\mathcal{F}$. See text for more details.

Additional works consider video synthesis based on one [51] or multiple layouts [46, 74], another video [8], or sound [10, 34, 73]. To account for the variety of potential controls, we leverage the flexibility of transformers, and propose an unifying approach to tackle all of these tasks.

# 3 Context-aware controllable video synthesis

## 3.1 Overview of the proposed approach

We consider video data, such that an individual frame $x$ is natively an element of $\mathcal{X} = \mathbb{R}^{H \times W \times 3}$, which can be encoded as some feature $z$ in $\mathcal{Z} = \mathbb{R}^{h \times w \times F}$ with reduced spatial resolution and increased number of channels. We assume that we are given $X_c$ in $\mathcal{X}^m$ corresponding to $m$ successive frames, and our goal is to *synthesize* a plausible representation of the following $n$ frames $X_s$ which lies in $\mathcal{X}^n$. Given $X_c$, we can compute the corresponding features $Z_c$ in $\mathcal{Z}^m$ using some encoder $\mathcal{E} : \mathcal{X} \to \mathcal{Z}$ on each frame individually, then use an autoregressive model (a transformer coupled with a quantization step in our case) to predict features $Z_s$ in $\mathcal{Z}^n$ formed by the $n$ features corresponding to time ticks $m + 1$ to $m + n$. (Note that the values of $m$ and $n$ can be arbitrary, using [by now] traditional masking and sliding window techniques.) These features can finally be converted one by one into the corresponding frames $X_s$ using some decoder $\mathcal{D} : \mathcal{Z} \to \mathcal{X}$.

The overall approach is illustrated by Figure 1. The use of a quantizer over a learned codebook in our implementation complicates the architecture a bit, but has several advantages, including the reuse of familiar NLP technology [68] and, perhaps more importantly, affording simple mechanisms for handling different types of inputs (from images to sound for example [9, 31]) and the intrinsically uncertain and multimodal nature of future prediction (by sampling different codebook elements according to their likelihood [30]). Concretely, this choice simply amounts to inserting in our architecture, right after the encoder $\mathcal{E}$, a nearest-neighbor quantizer $\mathcal{Q} : \mathbb{R}^F \to [\![1, q]\!]$ parameterized by an $F \times q$ codebook which, given the embedding $z$ of a frame, returns a $h \times w$ matrix of corresponding *tokens*, that is, the indices of the closest entry in the codebook for feature vector $z_{i,j}$ for all spatial locations $(i, j)$ in $[\![1, h]\!] \times [\![1, w]\!]$. We abuse notation, and identify $\mathcal{Q}$ with its parameterization by this codebook, so we can optimize over $\mathcal{Q}$ just as we optimize over $\mathcal{E}$ instead of naming explicitly its parameters in the rest of this presentation. An "un-quantizer" $\mathcal{U} : [\![1, q]\!] \to \mathbb{R}^F$, also parameterized (implicitly) by the codebook and associating with each token the corresponding entry of the codebook, is also inserted right before the decoder $\mathcal{D}$. In this setting, $\mathcal{T} : [\![1, q]\!]^{m \times h \times w} \to [\![1, q]\!]^{n \times h \times w}$ takes as input a sequence of tokens, and outputs the tokens for subsequent frames, each one of them chosen among $q$ possibilities as either the one with the highest score, or drawn randomly

from the top-$k$ scores to account for the multimodal nature of future forecasting (here $k \leq q$ is some predefined constant, see [30] for related approaches).

The elements of the architecture described so far are by now (individually) rather classical, with learnable, parametric functions $\mathcal{E}$, $\mathcal{D}$, $\mathcal{Q}$ and $\mathcal{T}$. Besides putting them all together, and as detailed in the rest of this section, we add several original elements: **(a)** The encoding/decoding scheme is improved by the use of two discriminators, $\Delta_i$ and $\Delta_t$ respectively, trained in an adversarial manner to ensure that the predicted frames are realistic and temporally consistent. **(b)** The context frames $X_c$ are themselves updated each time a new frame is predicted in an autoregressive manner (iteratively fill the sequence up to some predefined capacity, then shift to the left, forgetting the first frame and adding the latest synthetic one on the right). **(c)** The encoder and decoder are linked through a learnable *flow module* $\mathcal{F}$, allowing the context frames to guide the prediction of the synthetic ones. **(d)** Additional *control variables*, ranging from object trajectories to audio tracks, can be used in the form of sequence- or frame-level annotations to drive the synthesis by adding the corresponding tokens to the ones passed on to the autoregressive model $\mathcal{T}$.

We detail in this section the concrete components of the approach sketched above, including the autoencoder and quantizer architectures and their training procedure (Section 3.2), and the implementation of the autoregressive model by a transformer [68], illustrated in Figure 2, which we adapt to account for outside control signals (Section 3.3), once again with the corresponding procedure. Further architectural choices are also detailed in Appendix A.

### 3.2  First stage: training the context-aware autoencoder and the quantizer

**Architecture.**  $\mathcal{E}$ and $\mathcal{D}$ respectively decreases and increases the spatial resolution by using convolutional residual blocks [29], with $(r_k)_{k \in [\![1;K]\!]}$ the $K$ corresponding resolution levels ($r_k = h_k \times w_k$). It is common practice to augment, as in U-Net [55], the autoencoder with long skip connections between $\mathcal{E}$ and $\mathcal{D}$ to share information across the two models at these intermediate levels and escape the lossy compression bottleneck. Although we cannot apply this directly to our setting since information only flows through $\mathcal{D}$ for predicted timesteps, such skip connections can be established from the encoding stage of a context frame $x_c$ to the decoding stage of a new frame $x_s$ (resulting from features $z_s$). Similar mechanisms [15, 19] have been proposed in the past for video synthesis but they are only copying static background features from a single context frame. We follow works on semantic segmentation [23] and face frontalization [78] and use a flow module $\mathcal{F}$ to warp features and produce temporally consistent outputs despite motion. We extend this to multi-frame contexts, with significant performance gains and no additional parameter to be learned.

Concretely, let $e_c^k$ be features being encoded from $x_c$, and $d_s^k$ features being decoded from $z_s$ at a given intermediate resolution $r_k$. We first compute all intermediate context features $e_c^k$ by applying $\mathcal{E}$ to $x_c$. We then progressively decode features $d_s^k$ for the new frame from low ($k = 1$) to high resolution ($k = K$) by iterating over the following steps: **(a)** apply one decoding sub-module to get $d_s^k$ from $d_s^{k-1}$, **(b)** use $\mathcal{F}$ to refine the optical flow $f_c^k$ (in $\mathbb{R}^{2 \times r_k}$) which estimates the displacement field from $e_c^k$ to $d_s^k$ (as a proxy to the one from $x_c$ to $x_s$), and a fusion mask $m_c^k$ (in $\mathbb{R}^{1 \times r_k}$) which indicates the expected similarity between aligned features $e_c'^k = \mathrm{W}(e_c^k, f_c^k)$ and $d_s^k$ (also as a proxy for the one in image domain) with $\mathrm{W}$ corresponding to a standard warping operation, **(c)** use $e_c'^k$ and $m_c^k$ to update $d_s^k$ with context information (see update rule (1) below), **(d)** move to resolution level $r_{k+1}$ by going back to **(a)**. We note that $\mathcal{F}$ estimates $f_c^k$ and $m_c^k$ in a coarse-to-fine fashion by refining $f_c^{k-1}$ and $m_c^{k-1}$ (see Appendix A for further details). Temporal skip connections at a given resolution level $r_k$ are defined as the following in-place modification of $d_s^k$:

$$d_s^k = \sigma(m_c^k) \otimes d_s^k + (\mathbf{1} - \sigma(m_c^k)) \otimes e_c'^k, \tag{1}$$

where $\sigma$ is the Sigmoid function, and $\otimes$ the element-wise product. We note that update rule (1) is quite standard for warping and fusing two streams of spatial information [23, 28, 74]. For concrete implementation of $\mathcal{F}$, we build upon LiteFlowNet [32], an optical flow estimation model which also combines pyramidal extraction and progressive warping of features. For simple integration into our framework, we use features from $\mathcal{E}$ and $\mathcal{D}$ both in the mask and flow estimation, and in the update (1). This process readily generalizes to multi-frame contextual information (see Appendix D for details). Similar to spatial transformers [33], the warping operation $\mathrm{W}$ is differentiable. As a result, gradients from the training losses can backpropagate from $\mathcal{D}$ to $\mathcal{E}$ through $\mathcal{F}$. This allows end-to-end training of the autoencoder even with information from different timesteps in $\mathcal{E}$ and $\mathcal{D}$.

**Training procedure.** The global objective is the linear combination of four auxiliary ones:

$$\mathcal{L} = \lambda_q \mathcal{L}_q + \lambda_r \mathcal{L}_r + \lambda_a \mathcal{L}_a + \lambda_c \mathcal{L}_c, \tag{2}$$

namely a quantization loss ($\mathcal{L}_q$), a reconstruction loss ($\mathcal{L}_r$), an adversarial loss ($\mathcal{L}_a$), and a contextual loss ($\mathcal{L}_c$), detailed in the next paragraphs.

The codebook is trained by minimizing the reconstruction error between encoded features $z = \mathcal{E}(x)$ and quantized features $z_q = \mathcal{U}(\mathcal{Q}(z))$ (with notations introduced in Section 3.1):

$$\mathcal{L}_q(\mathcal{E}, \mathcal{Q}) = \| \operatorname{sg}(z) - z_q \|_2^2 + \beta \| \operatorname{sg}(z_q) - z \|_2^2, \tag{3}$$

where $\operatorname{sg}(.)$ is the stop gradient operation which constrains its operand to remain constant during backpropagation. The first part moves the codebook entries closer to the encoded features. The second part, known as the *commitment loss* [67], reverses the roles played by the two variables.

In regions with complex textures and high frequency details, local patterns shifted by a few pixels in $x$ and its reconstruction $\widehat{x} = \mathcal{D}(z_q)$ may result in large pixel-to-pixel errors, while being visually satisfactory. We thus define the recovery objective as the $L_1$ loss in both RGB space and between features from a VGG network [59] pretrained on ImageNet [14]:

$$\mathcal{L}_r(\mathcal{E}, \mathcal{Q}, \mathcal{F}, \mathcal{D}) = \|x - \widehat{x}\|_1 + \| \operatorname{VGG}(x) - \operatorname{VGG}(\widehat{x})\|_1. \tag{4}$$

To tackle cases where information cannot be recovered from context due to occlusion, and the compressed features are insufficient to create plausible reconstructions due to lossy compression, we supplement our architecture with an image discriminator $\Delta_i$, made of downsampling residual blocks [29], to encourage realistic outputs. $\Delta_i$ tries to distinguish real images from reconstructed ones ($\mathcal{L}_d$), while $\mathcal{E}$ and $\mathcal{D}$ fools $\Delta_i$ into assuming reconstructed images are as good as real ones ($\mathcal{L}_a$):

$$\mathcal{L}_d(\Delta_i) = \ln(1 + e^{\Delta_i(x)}) + \ln(1 + e^{-\Delta_i(\widehat{x})}), \quad (5) \qquad \mathcal{L}_a(\mathcal{E}, \mathcal{Q}, \mathcal{F}, \mathcal{D}) = \ln(1 + e^{\Delta_i(\widehat{x})}). \tag{6}$$

We employ a similar strategy on sequences of consecutive frames to improve the temporal consistency using a 3D temporal discriminator $\Delta_t$, a direct extension of 2D image discriminator $\Delta_i$.

The success of our method relies on accurate motion estimation in $\mathcal{F}$, a difficult task which benefits from self-supervision [35]. Therefore, we train the autoencoder with augmented views of the input frames as context. Custom augmentations functions $A : \mathcal{X} \to \mathcal{X}$ include: rotation, scaling, translation, elastic deformation, and combinations of these. Augmented views are obtained by warping $x$ by the suitable flow $a_c$: $A(x) = W(x, a_c)$, and the inverted flow $f_c$ from $A(x)$ to $x$ can be approximated.[1] We resort to flow inversion because directly reconstructing distorted views may encourage similar defects during inference. Moreover, we balance between self-recovery and context-recovery objectives by additionally applying a blurring function $B : \mathcal{X} \to \mathcal{X}$ and an occlusion mask $o_c$ to the augmented frames $x_c = o_c \otimes B(A(x))$. This augmentation strategy is illustrated in Appendix F. We define the contextual loss as:

$$\mathcal{L}_c(\mathcal{E}, \mathcal{Q}, \mathcal{F}, \mathcal{D}) = \|f_c - \widehat{f}_c\|_2^2 + \|o_c' - \sigma(\widehat{m}_c)\|_2^2, \tag{7}$$

where $o_c' = W(o_c, f_c)$, and $\widehat{f}_c$ and $\widehat{m}_c$ are the flow and mask estimated by $\mathcal{F}$. In practice, this loss is applied at intermediate resolution levels $r_k$ for improved training.

### 3.3 Second stage: predicting temporal dynamics with transformers

**Architecture.** We follow [20] and adopt an architecture similar to Image-GPT [9] for $\mathcal{T}$. Instead of modeling a single annotated frame as in [54], we design our model to account for sequences of $N$ such frames to allow prediction of temporal dynamics controlled by ancillary information in the form of video- and frame-level annotations. We have shown, in Section 3.2, how to represent a frame as a sequence of $h \times w$ tokens (indices in $[\![1, q]\!]$) through encoding and quantization. Similar strategies can be applied to cater to other types of data, with or without compression depending on their complexity, and, thereby, turn ancillary information into tokens as well. The capacity of the transformer (the maximum sequence length it can process) is $L = l_v + N * (l_f + h * w)$, where $l_v$ and $l_f$ are the size of video- and frame-level annotations respectively. The final layer of the model predicts, for every $i$ in $[\![1, L - 1]\!]$, a vector $\widehat{o}_i$ (of size $q$, the number of possible tokens) which scores the likelihood of the $i + 1^{\text{th}}$ token given all preceding ones.

---

[1]Our implementation of flow inversion approximation is detailed in Appendix C.

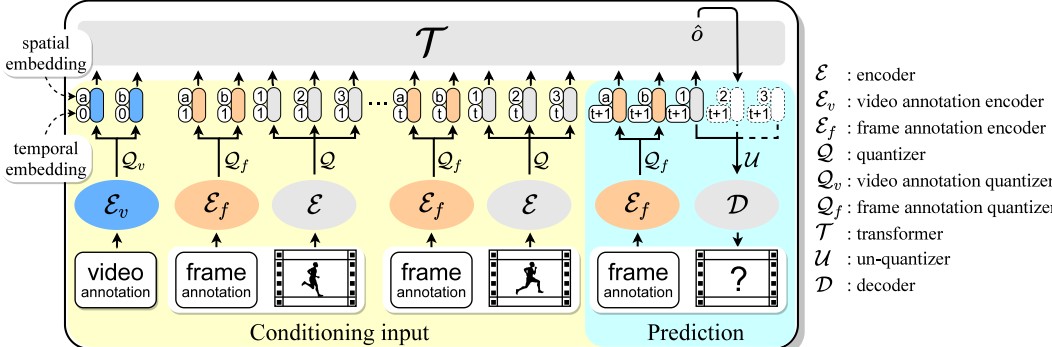

Figure 2: Illustration of the transformer to predict latent temporal dynamics with control-related data. For clarity, we omit the flow module $\mathcal{F}$ between $\mathcal{E}$ and $\mathcal{D}$. Input video-level annotation, frame-level annotations, and conditioning frames are encoded and quantized to form the initial token sequence, and mapped to corresponding embeddings. Subsequent frame tokens are obtained autoregressively with $\mathcal{T}$, while the ones corresponding to video- and frame-level annotations guide the prediction.

**Training procedure.** To learn the parameters of $\mathcal{T}$, we load a complete sequence ($N$ frames corresponding to $L$ tokens), and try to predict the $L-1$ last tokens based on the $L-1$ first ones. This is done by maximizing the log-likelihood of the data using the cross-entropy loss:

$$\mathcal{L}(\mathcal{T}) = -\sum_{i=2}^{L} \log \left( \frac{\exp(\langle \widehat{o}_{i-1}, e_{\tau(i)} \rangle)}{\sum_j \exp(\langle \widehat{o}_{i-1}, e_j \rangle)} \right), \tag{8}$$

where $\tau(i)$ is the $i^{\text{th}}$ ground-truth token, and $e_j$ a vector of 0's with a 1 in its $j^{\text{th}}$ coordinate.

**Inference.** During inference we use $\mathcal{T}$ to complete autoregressively an input sequence of tokens. To avoid known pitfalls (*e.g.*, repetitive synthesis) and allow diverse outcomes, we use top-$k$ sampling whereby the next token is randomly chosen among the $k$ most likely ones, weighted by their scores $\widehat{o}$. Although $\mathcal{T}$ processes sequences of $N$ annotated frames, we predict ones of arbitrary length by using a temporal sliding window. Tokens corresponding to video- and frame-level annotations are given as input to $\mathcal{T}$ (not predicted, even for future frames) to guide the synthesis. We show in the experiments how this control can translates into a variety of interesting tasks.

## 4  Experiments

We assess the merits of CCVS in the light of extensive experiments on various video synthesis tasks.

**Datasets.** BAIR Robot Pushing [19] consists of 43k training and 256 test videos of a robotic arm interacting with objects from a fixed viewpoint. A high-resolution version has recently been released. We manually annotate, in 500 frames, the $(x, y)$ location of the arm in image space to train a position estimator, which we use for state-conditioned synthesis. To account for real world scenarios, we evaluate on Kinetics-600 [5], a large and diverse action-recognition dataset with approximately 500K videos. We also test our method on AudioSet-Drums [25] for sound-conditioned synthesis on music performance, containing 6k and 1k video clips in train and test splits respectively. Other datasets and tasks are covered in Appendix E.

**Metrics.** We use the Fréchet video distance (FVD) [66] which measures the distribution gap between real and synthetic videos in the feature space of an Inception3D network [6] pretrained on Kientics-400 [40]. It estimates the visual quality and temporal consistency of samples as well as the diversity in unconditioned scenarios. In conditioned ones, we use another metric for diversity (DIV) which is the mean pixel-wise distance among synthetic trajectories conditioned on the same input. For near-deterministic motions (*e.g.*, in reconstructions, or constrained tasks), there is a one-to-one mapping between real video frames and synthetic ones, and we include pairwise image quality assessments: the structural similarity index measure (SSIM) [77] which evaluates a per-frame

Table 1: Ablation study of the autoencoder on BAIR ($256 \times 256$). We evaluate self- and context-recovery modules in different scenarios: synthesizing 16-frame videos from known compressed features ("Reconstruction"), by inferring compressed features with $\mathcal{T}$ given the real trajectory of the robotic arm ("State-conditioned"), or without the trajectory ("Pred." and "Unc."). The first real frame is used as initial context in all cases, except for "Unc." where it is synthesized by StyleGAN2 [38].

| Self-recovery | | | | Ctxt.-recovery | | | Reconstruction | | State-conditioned | | Pred. | Unc. |
|---|---|---|---|---|---|---|---|---|---|---|---|---|
| RGB | VGG | $\Delta_i$ | $\Delta_t$ | $\mathcal{F}$ | Sup. | Ctxt. | FVD↓ | PSNR↑ | FVD↓ | PSNR↑ | FVD↓ | FVD↓ |
| ✓ | | | | | | 0 | $1200_{\pm 6}$ | 20.0 | $1238_{\pm 24}$ | 17.8 | $1265_{\pm 22}$ | $1321_{\pm 10}$ |
| ✓ | ✓ | | | | | 0 | $700_{\pm 11}$ | 17.9 | $714_{\pm 5}$ | 17.4 | $704_{\pm 7}$ | $765_{\pm 12}$ |
| ✓ | ✓ | ✓ | | | | 0 | $323_{\pm 3}$ | 17.8 | $355_{\pm 5}$ | 16.6 | $377_{\pm 11}$ | $566_{\pm 22}$ |
| ✓ | ✓ | ✓ | ✓ | | | 0 | $389_{\pm 12}$ | 18.2 | $401_{\pm 4}$ | 16.9 | $407_{\pm 10}$ | $627_{\pm 9}$ |
| ✓ | ✓ | ✓ | ✓ | ✓ | | 1 | $98_{\pm 2}$ | 22.7 | $106_{\pm 2}$ | 22.2 | $142_{\pm 6}$ | $350_{\pm 11}$ |
| ✓ | ✓ | ✓ | ✓ | ✓ | ✓ | 1 | $87_{\pm 3}$ | 24.4 | $97_{\pm 4}$ | 22.1 | $128_{\pm 4}$ | $301_{\pm 10}$ |
| ✓ | ✓ | ✓ | ✓ | ✓ | ✓ | 8 | $62_{\pm 1}$ | 25.4 | $76_{\pm 3}$ | 22.7 | $109_{\pm 6}$ | $299_{\pm 4}$ |
| ✓ | ✓ | ✓ | ✓ | ✓ | ✓ | 15 | $60_{\pm 1}$ | 25.6 | $75_{\pm 2}$ | **22.8** | $110_{\pm 3}$ | $297_{\pm 7}$ |
| *Training longer* | | *(num. epochs* $\times 3$*)* | | | | | $\mathbf{45_{\pm 1}}$ | **26.8** | $\mathbf{67_{\pm 1}}$ | 22.3 | $\mathbf{100_{\pm 2}}$ | $\mathbf{293_{\pm 7}}$ |

"Sup.": self-supervision of $\mathcal{F}$;    "Ctxt.": number of context frames taken into account (in $\mathcal{F}$) when decoding current frame (in $\mathcal{D}$).

Table 2: Ablation study of the transformer on BAIR. We adopt notations and evaluation setups from Table 1.

| Architecture | | | Top-k | | State-conditioned | | Pred. | Unc. |
|---|---|---|---|---|---|---|---|---|
| Layer | Head | Dec. | Frame | State | FVD↓ | PSNR↑ | FVD↓ | FVD↓ |
| 6 | 4 | | 1 | 1 | $73_{\pm 2}$ | 23.0 | $281_{\pm 7}$ | $474_{\pm 17}$ |
| 12 | 8 | | 1 | 1 | $73_{\pm 3}$ | 23.1 | $262_{\pm 7}$ | $435_{\pm 8}$ |
| 24 | 16 | | 1 | 1 | $70_{\pm 3}$ | **23.3** | $331_{\pm 9}$ | $521_{\pm 19}$ |
| 24 | 16 | ✓ | 1 | 1 | $69_{\pm 2}$ | 23.2 | $321_{\pm 9}$ | $479_{\pm 34}$ |
| 24 | 16 | ✓ | 10 | 1 | $\mathbf{65_{\pm 2}}$ | 22.4 | $127_{\pm 7}$ | $308_{\pm 19}$ |
| 24 | 16 | ✓ | 100 | 1 | $67_{\pm 1}$ | 22.3 | $121_{\pm 2}$ | $314_{\pm 12}$ |
| 24 | 16 | ✓ | 100 | 10 | $67_{\pm 1}$ | 22.3 | $\mathbf{100_{\pm 2}}$ | $\mathbf{293_{\pm 7}}$ |

"Dec.": spatio-temporal decomposition of positional embeddings.

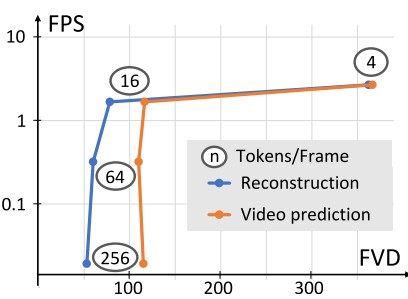

Figure 3: Quality and speed of synthesis vs. compression on a Nvidia V100 GPU.

conformity (combination of luminance, contrast and structure), and the peak signal-to-noise ratio (PSNR) which is directly related to the root mean squared error. For each metric, we compute the mean and standard deviation (std) over 5 evaluation runs (2 for Kinetics due to its voluminous 50k video test set). For clarity, the std value is shown only when it is greater than the reported precision.

**Training details.**    All our models are trained on 4 Nvidia V100 GPUs (32GB VRAM each), using ADAM [41] optimizer, for multiple 20 hour runs. We adapt the batch size to fill the memory available. We use a learning rate of $0.02$ to train the autoencoder, and exponential moving average [83] to obtain its final parameters. We use weighting factors $(1, 10, 1, 1)$ and $0.25$ for $(\lambda_q, \lambda_r, \lambda_a, \lambda_c)$ and $\beta$ in Equations (2) and (3) respectively. We use a learning rate of $10^{-5}$ to train the transformer.

### 4.1   Ablation study

We conduct an ablation study of CCVS to show the individual contribution of the key components of the proposed autoencoder (Table 1), transformer (Table 2), and the effect of compression (Figure 3).

First, we fix the transformer and observe the incremental improvements in synthesis quality when adding self- and context-recovery modules to the autoencoder (Table 1). In particular, $L_1$ loss in the feature space of a VGG net [59] produces sharper videos than using the same loss in the RGB space alone. Predicted frames using image and temporal discriminators ($\Delta_i$ and $\Delta_t$) display greater realism and temporal consistency. The flow module $\mathcal{F}$ significantly improves the performance on all metrics by allowing context frames to guide the reconstruction of synthetic ones. The self-supervision of $\mathcal{F}$, the use of larger context windows, and longer training times, further improve the quality of the synthesis. $\Delta_t$ seems to deteriorate the FVD at first, but when all modules are combined it improves FVD by almost a factor 2 as it encourages better temporal consistency in the presence of context.

Table 3: Future video prediction on BAIR ($64 \times 64$), synthesizing 16-frame videos given a few conditioning frames ("Cond."). We include some extensions of our method at higher resolution.

| Method | Cond. | FVD ↓ | Code Avail. | Memory, compute | |
|---|---|---|---|---|---|
| | | | | | $10^3$ $10^4$ $10^5$ → |
| MoCoGAN [65] | 4 | 503 | ✓ | 16GB, 23h[*] | GB×h |
| SVG-FP [15] | 2 | 315 | ✓ | 12GB, 6 to 24h[*] | |
| CDNA [21] | 2 | 297 | ✓ | 10GB, 20h[*] | |
| SV2P [2] | 2 | 263 | ✓ | 16GB, 24 to 48h[*] | |
| SRVP [22] | 2 | 181 | ✓ | 36GB, 168h[*] | |
| VideoFlow [42] | 3 | 131 | | 128GB, 336h[*] | |
| LVT [53] | 1 | $126_{\pm3}$ | ✓ | 128GB, 48h | |
| SAVP [44] | 2 | 116 | ✓ | 32GB, 144h | |
| DVD-GAN-FP [12] | 1 | 110 | | 2TB, 24h[*] | |
| Video Transformer (S) [79] | 1 | $106_{\pm3}$ | | 256GB, 33h[*] | |
| TriVD-GAN-FP [45] | 1 | 103 | | 1TB, 280h[*] | |
| *Low res.* CCVS (*ours*) | 1 | $99_{\pm2}$ | ✓ | 128GB, 40h | |
| Video Transformer (L) [79] | 1 | $94_{\pm2}$ | | 512GB, 336h[*] | |

| | | | SSIM ↑ ($t=8$) | SSIM ↑ ($t=15$) |
|---|---|---|---|---|
| *High res.*[**] CCVS (*ours*) | 1 | $80_{\pm3}$ | 0.729 | 0.683 |
| + end frame | 2 | $81_{\pm2}$ | 0.766 | 0.839 |
| + state | 1 | $50_{\pm1}$ | 0.885 | 0.863 |

[*]: value confirmed by authors;      [**]: training / inference / SSIM at $256 \times 256$, FVD at $64 \times 64$.

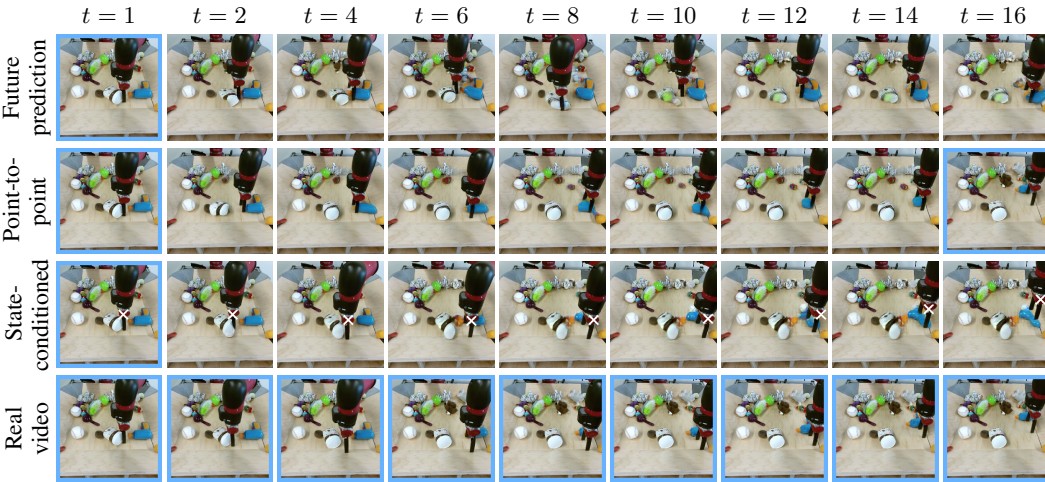

Figure 4: Qualitative samples for different types of control on BAIR ($256 \times 256$). Zoom in for details.

We fix the autoencoder, and compare different architectures and sampling strategies for the transformer (Table 2). Increasing the model capacity by adding layers and increasing expressivity (number of attention heads) along with a spatio-temporal decomposition of positional embeddings (shown in Figure 2, detailed in supplementary material) yields small improvements. Top-$k$ sampling is beneficial for stochastic tasks (video prediction and unconditional synthesis), whereas always selecting the most probable tokens results in little to no motion. Guiding temporal dynamics with state-conditioned synthesis reduces the advantage of sampling by narrowing down possible outcomes.

Finally, we explore the effect of compression (in terms of the number of tokens for each frame) on synthesis speed (FPS) and quality (FVD) of reconstructed and predicted videos (Figure 3). We use a compression of 64 tokens in our default setup since it gives the best FVD while retaining a reasonable speed. The critical drop of FPS for low compression ratios is due to the pairwise consideration of all input tokens in $\mathcal{T}$. Note that using a smaller temporal window in $\mathcal{T}$ may allow additional speed-ups.

Table 4: Future video prediction on Kinetics ($64\times64$) of 16-frame videos from 5 consecutive input frames.

| Method | FVD ↓ | GPU/TPU Mem. |
|---|---|---|
| LVT [53] | 225 | 128GB, 48h |
| Video Transformer [79] | $170_{\pm5}$ | 2TB, 336h* |
| DVD-GAN-FP [12] | $69_{\pm1}$ | 2TB, 144h* |
| CCVS (*ours*) | $55_{\pm1}$ | 128GB, 300h |
| TriVD-GAN-FP [45] | $26_{\pm1}$ | 16TB, 160h* |

*: value confirmed by authors.

Table 5: Codebook size on Kinetics.

| Size | Reconstruction | | | Pred. |
|---|---|---|---|---|
| | FVD ↓ | SSIM ↑ | PSNR ↑ | FVD ↓ |
| 1024 | $58_{\pm1}$ | 0.907 | 31.2 | $66_{\pm2}$ |
| 4096 | $54_{\pm1}$ | 0.917 | 31.6 | $64_{\pm1}$ |
| 16384 | $49_{\pm1}$ | 0.923 | 32.0 | $55_{\pm1}$ |
| 65536 | $45_{\pm1}$ | 0.928 | 32.2 | $61_{\pm1}$ |
| $\infty$ | $12_{\pm1}$ | 0.963 | 34.5 | $229_{\pm1}$ |

Table 6: Sound-conditional video synthesis on AudioSet-Drums ($64\times64$).

| Method | Cond. | Audio | SSIM↑ | | | PSNR↑ | | |
|---|---|---|---|---|---|---|---|---|
| | | | $t=16$ | $t=30$ | $t=45$ | $t=16$ | $t=30$ | $t=45$ |
| SVG-LP [15] | 15 | | $0.971_{\pm0.017}$ | $0.661_{\pm0.010}$ | $0.510_{\pm0.008}$ | $30.0_{\pm1.1}$ | $16.6_{\pm0.3}$ | $13.5_{\pm0.1}$ |
| Vougioukas *et al.* [73] | 15 | ✓ | $0.940_{\pm0.017}$ | $0.904_{\pm0.007}$ | $0.896_{\pm0.015}$ | $26.2_{\pm1.0}$ | $23.8_{\pm0.2}$ | $23.3_{\pm0.3}$ |
| Sound2Sight [10] | 15 | ✓ | $0.984_{\pm0.009}$ | $0.954_{\pm0.007}$ | $\mathbf{0.947}_{\pm0.007}$ | $33.2_{\pm0.1}$ | $27.9_{\pm0.5}$ | $27.0_{\pm0.3}$ |
| CCVS (*ours*) | 15 | ✓ | $\mathbf{0.987}_{\pm0.001}$ | $\mathbf{0.956}_{\pm0.006}$ | $0.945_{\pm0.008}$ | $\mathbf{33.7}_{\pm0.4}$ | $\mathbf{28.4}_{\pm0.6}$ | $\mathbf{27.3}_{\pm0.5}$ |

## 4.2 Quantitative and qualitative studies

**BAIR.** For future video prediction on BAIR (Table 3), CCVS trained at $64\times64$ resolution (*low res.*) is on par with the best method (L-size version of [79]), but requires much less computing resources, and outperforms [79] under similar resources. We also propose *high res.* CCVS which is not strictly comparable to the prior arts as we use $256\times256$ image resolution for training and test, and resize the synthesized frames to $64\times64$ for computing FVD. However, using this variant demonstrates the performance gains that can arise by scaling CCVS. We additionally address point-to-point synthesis (with the end frame as a video-level annotation) and state-conditioned synthesis (with the estimated 2D position of the arm as a frame-level annotation). Point-to-point synthesis is more difficult than video prediction: Not only does it requires realistic video continuations, but also ones which explain the end position of all visible objects. Hence, FVD score is constant despite the additional input. Still, this yields better SSIM for mid-point and one-before-last frames. State-conditioned synthesis improves on FVD and mid-synthesis SSIM as motion becomes near-deterministic. Some synthetic frames for these tasks are shown in Figure 4. CCVS creates plausible high-quality videos in various settings, and true interactions with objects compared to previous attempts [48] at the same resolution.

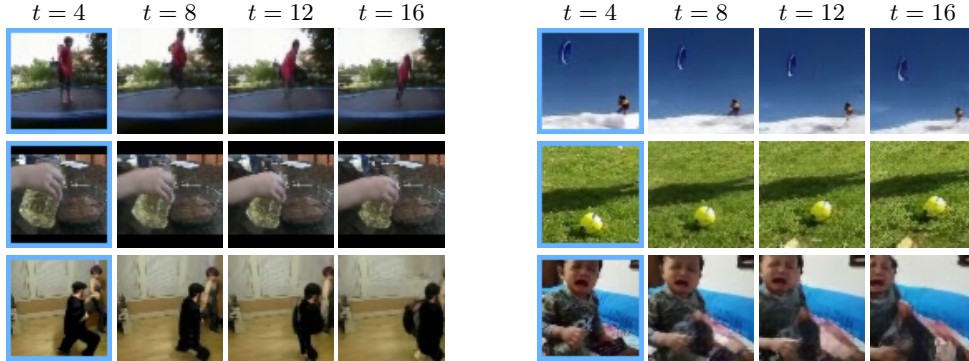

Figure 5: Qualitative samples on future video prediction on Kinetics ($64 \times 64$). Zoom in for details.

**Kinetics.** CCVS ranks second on Kinetics video prediction benchmark. Kinetics contains more diversity than BAIR, and the reconstruction is thus more difficult. A solution [54] is to increase the codebook size (Table 5) but it stops translating into better prediction FVD at some point. We also try removing the quantization step (equivalent to an infinite codebook), and directly regressing latent features with $\mathcal{T}$ (instead of ranking the likelihood of possible tokens). It allows better reconstructions, yet the prediction FVD is high. Figure 5 shows examples of synthetic continuations conditioned on

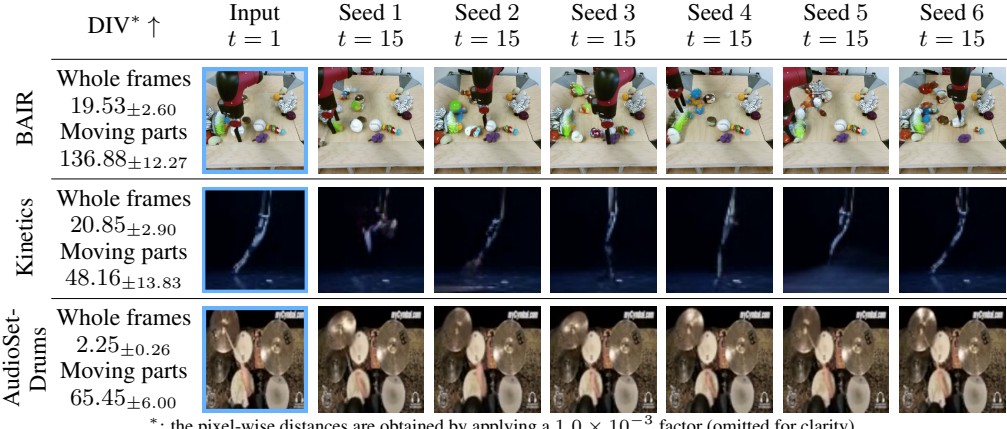

|  | DIV* ↑ | Input $t=1$ | Seed 1 $t=15$ | Seed 2 $t=15$ | Seed 3 $t=15$ | Seed 4 $t=15$ | Seed 5 $t=15$ | Seed 6 $t=15$ |
|---|---|---|---|---|---|---|---|---|
| BAIR | Whole frames $19.53_{\pm2.60}$ Moving parts $136.88_{\pm12.27}$ | | | | | | | |
| Kinetics | Whole frames $20.85_{\pm2.90}$ Moving parts $48.16_{\pm13.83}$ | | | | | | | |
| AudioSet-Drums | Whole frames $2.25_{\pm0.26}$ Moving parts $65.45_{\pm6.00}$ | | | | | | | |

*: the pixel-wise distances are obtained by applying a $1.0 \times 10^{-3}$ factor (omitted for clarity).

Figure 6: Diversity results for conditioned scenarios. The diversity metric (DIV) measures the mean pixel-wise distance among 10 15-frame synthetic trajectories conditioned on the same frame. We report the mean and std over 100 runs: on whole frames as in [82], or moving parts only by masking static regions (optical flow magnitude between consecutive frames $< 20\%$ of the max magnitude) as in [69]. We show, for the same input, the $15^{\text{th}}$ frame of various randomly seeded synthetic trajectories.

5-frame unseen test sequences. CCVS produces realistic and temporally coherent outputs which display various types of motion (e.g., body, hand, camera).

**AudioSet-Drums.** CCVS achieves top performance on sound-conditioned video synthesis on AudioSet Drums (Table 6). Figure 6 shows quantitative and qualitative insights on the diversity of the synthetic trajectories conditioned on the same input for the three datasets. The diversity metrics (DIV) computed on whole frames is lower on AudioSet Drums than on the other two datasets. This is explained by the fact that motion is quite repetitive and involves a limited portion of the frame. The same metric on moving parts only, and the end position of the drummer's hand and upper left cymbal in qualitative samples, show the diversity of synthetic trajectories. An ablation of CCVS with/without audio guidance as well as more qualitative results on diversity can be found in Appendix E.

## 5   Discussion

CCVS is on par or better than the state-of-the-art on standard benchmarks, uses less computational resources, and scales to high resolution. Training neural networks is environmentally costly, due to the carbon footprint to power processing hardware [17, 61]. Methods sparing GPU-hours like ours are crucial to make AI less polluting [43, 61], and move from a "Red" to a "Green" AI [58]. Future work will include exploring new codebook strategies and synthesis guided by textual information.

**Limitations.** CCVS uses a complex architecture and a two-stage training strategy. Simplification of both is an interesting direction for improving the method. Moreover, CCVS lacks global regularization of motion (flow computed on pairs of timesteps), and its efficiency relies on recycling context information such that synthesizing content from scratch (i.e., no input frame given) remains difficult.

**Broader impact.** The increased accessibility and the many controls CCVS offers could accelerate the emergence of questionable applications, such as "deepfakes" (e.g., a video created from someone's picture and an arbitrary audio) which could lead to harassment, defamation, or dissemination of fake news. On top of current efforts to automate their detection [49], it remains our responsibility to grow awareness of these possible misuses. Despite these worrying aspects, our contribution has plenty of positive applications which outweigh the potential ethical harms. Our efficient compression scheme is a step in the direction of real-time solutions: e.g., enhancing human-robot interactions, or improving the safety of self-driving cars by predicting the trajectories of people and vehicles nearby.

## Acknowledgements

This work was granted access to the HPC resources of IDRIS under the allocation 2020-AD011012227 made by GENCI. It was funded in part by the French government under management of Agence Nationale de la Recherche as part of the "Investissements d'avenir" program, reference ANR-19-P3IA-0001 (PRAIRIE 3IA Institute). JP was supported in part by the Louis Vuitton/ENS chair in artificial intelligence and the Inria/NYU collaboration. We thank the reviewers for useful comments.

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
