# A  Detailed architecture design

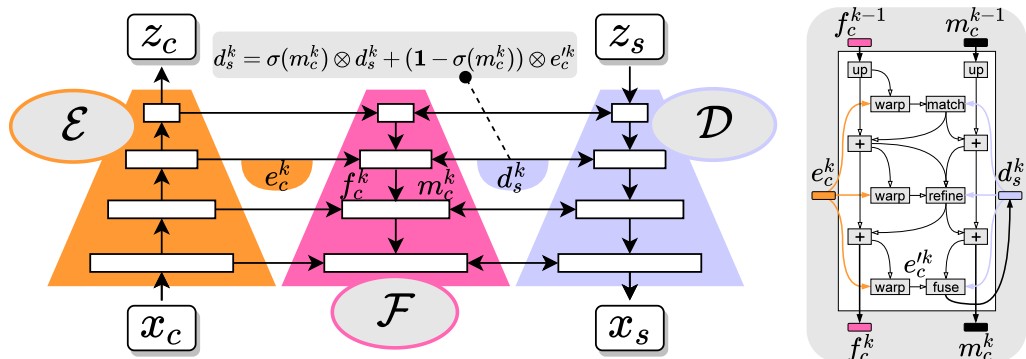

$$d_s^k = \sigma(m_c^k) \otimes d_s^k + (\mathbf{1} - \sigma(m_c^k)) \otimes e_c'^k$$

(1) Conversion from image to feature space ($\mathcal{E}$) and vice versa ($\mathcal{D}$). In the later case, context frames help producing new ones via $\mathcal{F}$.

(2) Computations in $\mathcal{F}$ at one resolution level.

Figure A1: In (1), illustration of the encoding ($\mathcal{E}$) and decoding ($\mathcal{D}$) architectures, and the interactions between the two thanks to the progressive optical flow and fusion mask estimation operator ($\mathcal{F}$). As opposed to $\mathcal{E}$, $\mathcal{D}$ processes features from low to high spatial resolution. To allow feature updates from context frame $x_c$ in the decoding phase of a latent embedding $z_s$, we first compute encoded features $e_c^k$ for all intermediate resolution levels $(r_k)_{k\in[\![1,K]\!]}$ by applying $\mathcal{E}$ to $x_c$. We then decode $x_s = \mathcal{D}(z_s)$ from low ($k = 1$) to high resolution ($k = K$) by iterating over the following steps: **(a)** apply one decoding sub-module to get $d_s^k$ from $d_s^{k-1}$, **(b)** use $d_s^k$ and $e_c^k$ (at corresponding levels) to estimate the flow $f_c^k$ from $e_c^k$ to $d_s^k$ and a fusion mask $m_c^k$, **(c)** use $f_c^k$ and $m_c^k$ to update $d_s^k$ with $e_c^k$ (see update rule), and **(d)** move to the next resolution level by going back to **(a)**. The residual computation of $f_c^k$ and $m_c^k$ from $e_c^k$, $d_s^k$, $f_c^{k-1}$ and $m_c^{k-1}$ is shown in (2) and detailed in the text.

We describe the implementation choices for the different functional blocks which compose CCVS. Note that different parts of the framework may benefit from future advances in various research areas: autoregressive modeling, image synthesis, codebook learning and optical flow estimation for transformer $\mathcal{T}$, decoder $\mathcal{D}$, quantizer $\mathcal{Q}$ and flow module $\mathcal{F}$ respectively.

**Encoder and decoder.**    Both the encoder $\mathcal{E}$ and the decoder $\mathcal{D}$, illustrated in Figure A1-(1), use a sequence of $K$ modules handling information at different intermediate resolution levels $(r_k)_{k\in[\![1;K]\!]}$ (with reverse ordering from $\mathcal{E}$ to $\mathcal{D}$). Each module is implemented by a residual block [29] which increases or decreases the spatial resolution by a factor 2. In practice, we use the succession of two $3 \times 3$ 2D convolutions with LeakyReLU activations and a skip connection in the form of a $1 \times 1$ convolution.

**Flow module.**    The flow module $\mathcal{F}$, also illustrated in Figure A1-(1), allows the sharing of information between the encoding and decoding stages across different timesteps. Computations for one resolution level $r_k$ are detailed in Figure A1-(2): The "match" operation computes the correlation between features in $d_s^k$ and its closest spatial neighbours in $e_c^k$. We thus obtain a 3D cost volume of size $N \times h_k \times w_k$, where $N$ is the number of neighbours considered and $(h_k \times w_k) = r_k$ is the spatial resolution. We apply 2D convolutions to this cost volume to deduce the residual flow and mask. The "refine" operation is also a residual update of $f_c^k$ and $m_c^k$. It is made of 2D convolutions and takes as input the concatenation of features, mask and flow. Finally, the "fuse" operation update features from $\mathcal{D}$ with information from the context in $\mathcal{E}$ according to the update rule given in Figure A1-(1) and detailed in Section 3.2. For these operations, we take our inspiration from LiteFlowNet [32] which we adapt to handle not only the optical flow but also the proposed fusion mask. An extension to multi-frame context for this temporal skip connection mechanism is proposed in Appendix D.

**Image and temporal discriminators.**    The architecture of the image discriminator $\Delta_i$ is similar to that of StyleGAN2 [38], and the temporal discriminator $\Delta_t$ is essentially the same, with 3D instead of 2D convolutions to account for the temporal dimension. To avoid the propagation of errors when

Table B1: Reconstruction of 16-frame videos from compressed features ($z_s$) and the first frame on BAIR ($256 \times 256$). Features $z_s$ are either fixed (copied from the first timestep for subsequent timesteps) or dynamic (the true ones, obtained by applying $\mathcal{E}$ frame-by-frame to the video).

| $z_s$ | FVD$\downarrow$ | PSNR$\uparrow$ |
|---|---|---|
| Fixed | $556_{\pm 12}$ | 18.9 |
| Dynamic | $45_{\pm 1}$ | 26.8 |

decoding frames autoregressively during inference (see the "shift-and-add" module introduced for this purpose in Figure 1), we mimic this iterative process at train time: decode frames one by one, then encode them back to obtain context features for the next steps. The resulting synthetic videos are fed to $\Delta_t$. To avoid excessive memory consumption due to the recursive encoding / decoding operations, we only keep the gradients for the context features from the immediate preceding frame.

**Quantizer.** The implementation of the quantizer $\mathcal{Q}$ is straightforward since it is just using a codebook of learnable embeddings. Yet, the non-differentiable nature of this operation prevents gradients from training losses (introduced in Section 3.2) to backpropagate through $\mathcal{U}$ and $\mathcal{Q}$, and from $\mathcal{D}$ to $\mathcal{E}$ at the compression bottleneck. To solve this in practice, we use a straight-through gradient estimator [3] which ignores the quantization step during backward-pass.

**Transformer.** The transformer $\mathcal{T}$ is described in Section 3.3. Its input is obtained by transforming a sequence of tokens into a 2D matrix of size $L \times D$. For that, $\mathcal{T}$ associates with each possible token and each of the $L$ positions in the sequence a learnable embedding of size $D$. To leverage the prior information that successive fixed-sized portions of the sequence represent annotated frames, we propose a spatio-temporal decomposition of the positional embeddings with a term indicating the timestep, and another one indicating the position within the frame-related portion. This is illustrated in Figure 2. The input matrix is computed by adding the token embedding and the positional embedding for each token of the sequence. $\mathcal{T}$ thus consists in the succession of multi-head self-attention layers, and position-wise fully connected layers.

**Position estimator.** In the experiments (Section 4), we mention that we manually annotated the $(x, y)$ location of the robotic arm (more precisely, its gripper) in 500 frames of the BAIR dataset. We use these to train a position estimator so that we can extract the trajectory of the arm from any new video (and use this trajectory to guide the state-conditioned synthesis of new videos). To facilitate training and accelerate inference, we design our position estimator to exploit latent features (obtained from images with $\mathcal{E}$) and to map them to the $(x, y)$ position of the arm. We use a simple architecture with a few downsampling convolutional layers and a fully connected layer to output the estimated position. The position estimator is trained by regressing the target 2D coordinates by minimizing the corresponding mean squared error.

**Sound autoencoder.** To allow multimodal synthesis of video and sound, we first compute the short-time Fourier transform (STFT) of the raw audio to obtain a time-frequency representation of the sound. We then employ an encoding / quantization / decoding strategy similar to that used to construct $\mathcal{E}$, $\mathcal{Q}$ and $\mathcal{D}$ for video frames (without $\mathcal{F}$ in this case) to reduce dimension and obtain corresponding tokens for audio. These tokens are used as frame-level annotations to guide the synthesis, just as in state-conditioned synthesis.

# B  Role of the flow module

Looking at Figure A1, one may wonder whether the flow module $\mathcal{F}$ alone is sufficient for predicting features for future frames which would result in $z_s$ potentially being completely ignored. This would only apply if a frame was fully determined from the preceding one, yet, this is not the case due to the inherent stochastic nature of future prediction in the considered datasets. We have conducted a simple experiment to demonstrate that features $z_s$ are actually used and that they actively drive the dynamics of the scene: We have repeated the "reconstruction" experiment from Table 1 for which we compare the final model (last line) with an ablation with fixed $z_s$ (*i.e.*, fixing $z_s = z_c$ for all predicted timesteps). Results are available in Table B1.

We observe that the reconstruction performance is much poorer when we force $z_s$ to remain constant for all timesteps. Qualitatively, looking at the synthesized videos, we see that they do not exhibit any motion, as one would expect when fixing the encoded features. Moreover, thanks to the fusion masks estimated by the flow module $\mathcal{F}$ and used in Eq. (1), it is possible to observe how much information comes from $z_s$ and from $\mathcal{F}$ respectively when generating $x_s$. Figure F1 shows some examples of estimated masks on Cityscapes. Those are white when the source is $z_s$, and black when it is context through $\mathcal{F}$. In practice, we see that they are white when the context is occluded and mostly grey in other regions. Thus, final videos are as much the result of context warping than direct decoding in non-occluded regions for this dataset. We observe similar behaviours on other datasets.

## C  Flow inversion approximation

For affine transformations, the exact inverse flow can be determined analytically. This is not the case for elastic deformations for which we propose to approximate this inverse flow $f \in \mathbb{R}^{2 \times H \times W}$ from an input flow $g \in \mathbb{R}^{2 \times H \times W}$ (where $H \times W$ is the image spatial resolution). Here, $g$ maps cells from a grid $[\![1; H]\!] \times [\![1; W]\!]$ to cells from the same grid (we ignore cells for which $g$ points out of the grid since they do not help in computing $f$). Such a mapping is not surjective, that is, all the cells in the grid are not necessarily reached. We compute the pixel-accurate inversion in cells for which there exists a direct mapping, and approximate others by iterative interpolation. That is, we keep track of inverted cells with a completion mask $c$ which we update with the closest cells in cardinal directions at every step. For the elastic deformations that we use when training CCVS, around 80% of the grid can be directly inverted in average, which is enough to accurately reconstruct the missing cells by interpolation.

---

**Algorithm C1:** Flow inversion approximation.

---

**Data:** Flow $g \in \mathbb{R}^{2 \times H \times W}$
**Result:** Approximation of inverted flow $f \in \mathbb{R}^{2 \times H \times W}$
```
/* initialization of flow and cell-completion mask                  */
```
$f = \mathbf{0} \in \mathbb{R}^{2 \times H \times W}$;
$c = \mathbf{0} \in \{0; 1\}^{H \times W}$;
```
/* invert flow in cells for which there exists a direct mapping     */
```
**for** $(h, w) \in [\![1; H]\!] \times [\![1; W]\!]$ **do**
    $dh = g_{0,h,w}$;
    $dw = g_{1,h,w}$;
    $h' = \text{round}(h + dh)$;
    $w' = \text{round}(w + dw)$;
    **if** $(h', w') \in [\![1; H]\!] \times [\![1; W]\!]$ **then**
        $f_{0,h',w'} = -dh$;
        $f_{1,h',w'} = -dw$;
        $c_{h',w'} = 1$;
    **end**
**end**
```
/* fill empty cells iteratively by interpolating neighbours' flow    */
```
**while** $\exists (h, w) \in [\![1; H]\!] \times [\![1; W]\!] \mid \sim c_{h,w}$ **do**
    $c' = \text{D}(c) \wedge \sim c$;
    $i = \text{B}(f)$;
    $w = \text{B}(c)$;
    **for** $(h, w) \in [\![1; H]\!] \times [\![1; W]\!] \mid c'_{h,w}$ **do**
        $f_{0,h,w} = i_{0,h,w} / w_{h,w}$;
        $f_{1,h,w} = i_{1,h,w} / w_{h,w}$;
    **end**
    $c = c \vee c'$;
**end**

---

The blurring function $\text{B} : \mathbb{R}^{H \times W} \rightarrow \mathbb{R}^{H \times W}$ is a 2D convolution with $3 \times 3$ Gaussian kernel (and 0-valued 1-sized padding to preserve the spatial resolution), and interpolates flow values from neighbouring cells weighted by the their proximity; Dilation function $\text{D} : \{0; 1\}^{H \times W} \rightarrow \{0; 1\}^{H \times W}$

propagates 1-valued cells in a 2D boolean mask according to the cardinal directions (up, down, left, right); $\wedge$, $\vee$ and $\sim$ are notations which designate logical *AND*, *OR* and *NOT* respectively.

We note that an alternative to flow inversion is to use the flow module $\mathcal{F}$ to compute the backward flow. This requires an extra forward pass in the flow module to compute both forward and backward flows and would result in additional GPU computations. Not to slow down training we prefer to run flow inversion on parallelized CPU processes as part of data loading. While both alternatives take approximately the same amount of time (0.15s on a GPU for backward flow estimation for a batch of 16 images at resolution $256 \times 256$, 0.12s on parallelized CPU for flow inversion for the same input) flow inversion is advantageous because, in our setup, GPU consumption is the limiting factor for speed (fully utilized GPUs, data loading on CPUs can run in the background).

## D    Extension of the flow module to multi-frame context

We extend the flow-module (introduced in Section 3.2) which aims at reusing contextual information from single- to multi-frame context. We thus consider a context $(x_i)_{i \in [\![1;c]\!]}$ consisting of $c$ frames. Just as before, for a given level $r_k$ and $i$ in $[\![1;c]\!]$, we use $\mathcal{F}$ to compute the corresponding fusion mask $m_i^k$ and optical flow $f_i^k$ between intermediate encoded features $e_i^k$ (from context frame $x_i$) and the decoded features $d_s^k$ (that we wish to update). We recall that fusion masks $m_i^k$ handle occlusion by indicating for each spatial location the relevance of warped context features $e_i'^k = \mathrm{W}(e_i^k, f_i^k)$, that is, whether features $e_i'^k$ correspond to $d_s^k$ at that location in terms of content, or not. Hence, we can derive from the fusion masks a confidence score $s_i^k$ defined as the context-wise normalization of features relevance, and use it to aggregate all the information that can be recovered from context by applying a weighted average:

$$s_i^k = (\mathbf{1} - \sigma(m_i^k)) \oslash \left( \sum_{j=1}^{c} (\mathbf{1} - \sigma(m_j^k)) \right), \qquad \begin{cases} m_a^k = \sum_{i=1}^{c} s_i^k \otimes m_i^k \\ e_a'^k = \sum_{i=1}^{c} s_i^k \otimes e_i'^k \end{cases}. \qquad (9)$$

where $\oslash$ represents the element-wise division. In other words, given a spatial location, $m_a^k$ is an estimate for the likelihood that the content of $d_s^k$ matches any of the $c$ context frames, and $e_a'^k$ is used to reconstruct this content faithfully by weighting proposals from the context by their relevance. This brings us back to the problem setup of Section 3.2, so that the aggregated fusion mask $m_a^k$ and warped features $e_a'^k$ can be used to update $d_s^k$, just as in single-frame fusion. The key benefits of this extension are the use of redundancy within context to better reconstruct the current frame, and the greater robustness to occlusion as different views are combined. This is demonstrated by the ablation study of the autoencoder in Table 1.

## E    Additional results

Table D1: Unconditional video synthesis of 16-frame videos on UCF-101 ($128 \times 128$).

| Method | IS $\uparrow$ | FVD $\downarrow$ |
|---|---|---|
| StyleGAN2 [38] (repeat $\times 16$) | $17.98_{\pm.12}$ | $990_{\pm33}$ |
| Real frame (repeat $\times 16$) | $28.41_{\pm.11}$ | $838_{\pm27}$ |
| VGAN [71] | $8.31_{\pm.09}$ | - |
| TGAN [56] | $11.85_{\pm.07}$ | - |
| MoCoGAN [65] | $12.42_{\pm.07}$ | - |
| ProgressiveVGAN [1] | $14.56_{\pm.05}$ | - |
| LDVD-GAN [36] | $22.91_{\pm.19}$ | - |
| TGANv2 [57] | $26.60_{\pm.47}$ | $1209_{\pm28}$ |
| DVD-GAN [12] | $27.38_{\pm.53}$ | - |
| MoCoGAN-HD [63] | $33.95_{\pm.25}$ | $700_{\pm24}$ |
| StyleGAN2 [38] + CCVS (*ours*) | $24.47_{\pm.13}$ | $386_{\pm15}$ |
| Real frame + CCVS (*ours*) | $41.37_{\pm.39}$ | $389_{\pm14}$ |

**Unconditional synthesis.**    We apply CCVS to unconditional video synthesis (*i.e.*, the production of new videos without prior information about their content) on UCF-101 [60], an action-recognition dataset of 101 categories for a total of approximately 13k video clips. Our approach leverages recent

Table D2: Layout-conditioned video synthesis on Cityscapes ($256 \times 512$).

| Method | Cond. | Layout | SSIM↑ | | | PSNR↑ | | |
|--------|-------|--------|-------|--------|--------|--------|--------|--------|
| | | | $t = 10$ | $t = 20$ | $t = 30$ | $t = 10$ | $t = 20$ | $t = 30$ |
| CCVS (*ours*) | 3 | | $0.750_{\pm 0.002}$ | $0.639_{\pm 0.003}$ | $0.582_{\pm 0.005}$ | $20.7_{\pm 0.1}$ | $18.8_{\pm 0.1}$ | $17.9_{\pm 0.1}$ |
| CCVS (*ours*) | 3 | ✓ | $\mathbf{0.783}_{\pm 0.002}$ | $0.690_{\pm 0.003}$ | $0.640_{\pm 0.005}$ | $21.5_{\pm 0.1}$ | $19.6_{\pm 0.1}$ | $18.7_{\pm 0.1}$ |
| CCVS + [52] (*ours*) | 3 | ✓ | $0.776_{\pm 0.002}$ | $\mathbf{0.742}_{\pm 0.002}$ | $\mathbf{0.723}_{\pm 0.005}$ | $\mathbf{22.1}_{\pm 0.1}$ | $\mathbf{20.6}_{\pm 0.1}$ | $\mathbf{19.9}_{\pm 0.1}$ |

progress made in image synthesis by using StyleGAN2 [38] to produce the first frame. CCVS is able to improve over the state of the art on the FVD metric (computed on 2048 videos, under the same evaluation process as related approaches), but has an average performance ($24.47$) on the inception score (IS) [57] (computed on 10k videos, again with the appropriate protocol). IS measures the coverage of the different categories, and whether one of these is clearly identifiable in each of the synthetic videos, through the lens of the Softmax scores of a C3D network [64] trained for action recognition on the Sports-1M dataset [37] and fine-tuned on UCF-101. We also evaluate the performance of CCVS when considering a perfect image synthesizer (or one that has overfitted the training data) by using a real frame instead of a synthetic one to launch the prediction. This yields similar FVD but considerable improvements on IS ($41.37$). Note that CCVS favorably contribute to IS in this setting because videos made from the same real frames (by repeating them along the temporal axis) have much lower IS ($28.41$). A plausible explanation why MoCoGAN-HD [63] achieves a good IS score is due to it sampling multiple times from StyleGAN2 [38] (as opposed to once in our case, we use the same pretrained model). Indeed, MoCoGAN-HD synthesizes videos by finding a trajectory in the latent space of StyleGAN2. In this case, it may be easier to recognize a category because different modes can be interpolated (greater IS) but the outcome is less temporally realistic than ours (FVD near the one of still videos for MoCoGAN-HD). Future work will include trying to reconciliate both approaches.

**Layout-conditioned synthesis.** We explore here layout-conditioned synthesis where one controls the semantic structure of new frames by attributing a class to each of their spatial locations. To this end, we apply CCVS on image and layout sequence pairs on the Cityscapes dataset [13] which contains 3475 and 1525 30-frame sequences for train and test respectively. A single layout is annotated for each sequence, and we use a pretrained segmentation model [62] to obtain the remaining ones. Layouts are used as frame-level annotations for processing in $\mathcal{T}$. They are encoded, quantized, and decoded just as it is done for images. To save memory consumption during the decoding stage and ensure greater consistency between the two, frame and layout are decoded simultaneously with a single decoder $\mathcal{D}$ (we just supplement $\mathcal{D}$ with another prediction head to output layout masks). We show some qualitative results of CCVS on the synthesis of 30-frame videos given 3 starting frames and layouts for all timesteps in Figure D1. Synthetic videos closely follow the target semantic structure, remain temporally consistent, and can handle complex motions. We repeat this synthesis process in Figure D2, this time without layouts at subsequent timesteps. CCVS forecasts the semantic evolution of the scene on its own, predict corresponding layouts, and translate these into high-quality images. Both approaches (with or without layout at subsequent timesteps) are compared in Table D2. Having this additional information allows to produce videos which are closer to the real ones. A common architectural choice in layout-conditioned synthesis is to replace ResNet blocks [29] by SPADE blocks [52] to strengthen the layout guidance in the decoding stage [46]. Quantitatively, we see that it significantly improves the fidelity of the synthesized videos to the real ones in terms of SSIM and PSNR. Qualitatively, side-by-side reconstructions with and without SPADE in Figure D3 show that the flow module allows temporally consistent textures (similar to the 3D world model used in [46] but without being limited to static objects here) while SPADE enhances the compliance to the semantic structure, especially for small fast-moving objects.

**Sound ablation study.** A simple ablation of CCVS on AudioSet Drum [25] is presented in Table D3. It shows that, as time goes by, the performance gap between unimodal and multimodal synthesis (when considering audio) increases. However, the improvement on account of the audio information seems relatively small compared to the overall performance. We suspect this is due to a minor part of the scene being animated (the drummer upper body) and to relatively fast motion (it is difficult to strictly minimize the image-to-image error between real and synthetic videos in this case).

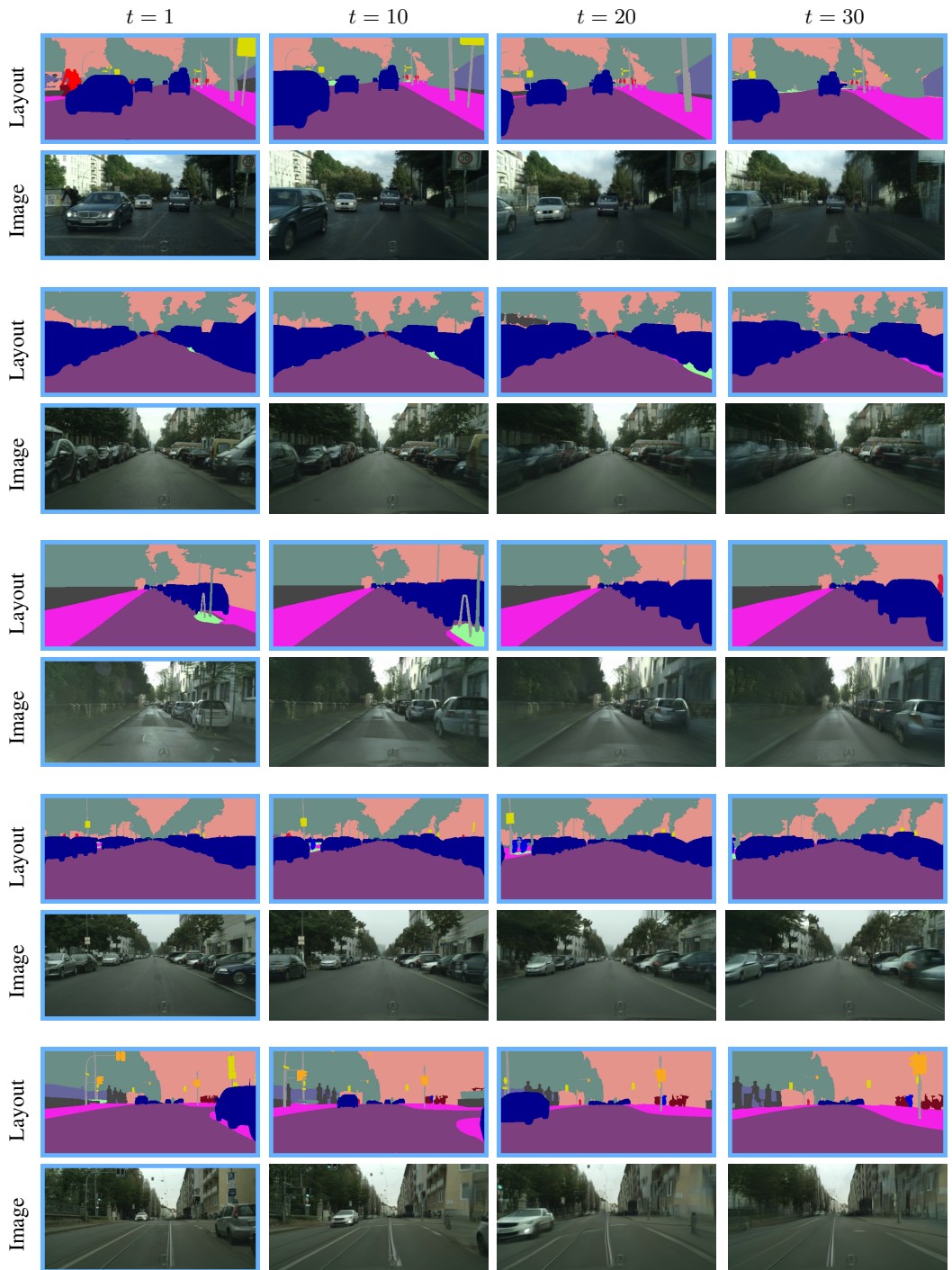

Figure D1: Layout-conditioned synthesis of 30-frame videos given 3 starting frames and layouts for all timesteps on Cityscapes ($256 \times 512$). We show different samples of real layout and synthetic image pairs at intermediate timesteps. We see that synthesized frames follow both the target semantic structure specified by the layouts and the texture extracted from the conditioning frames.

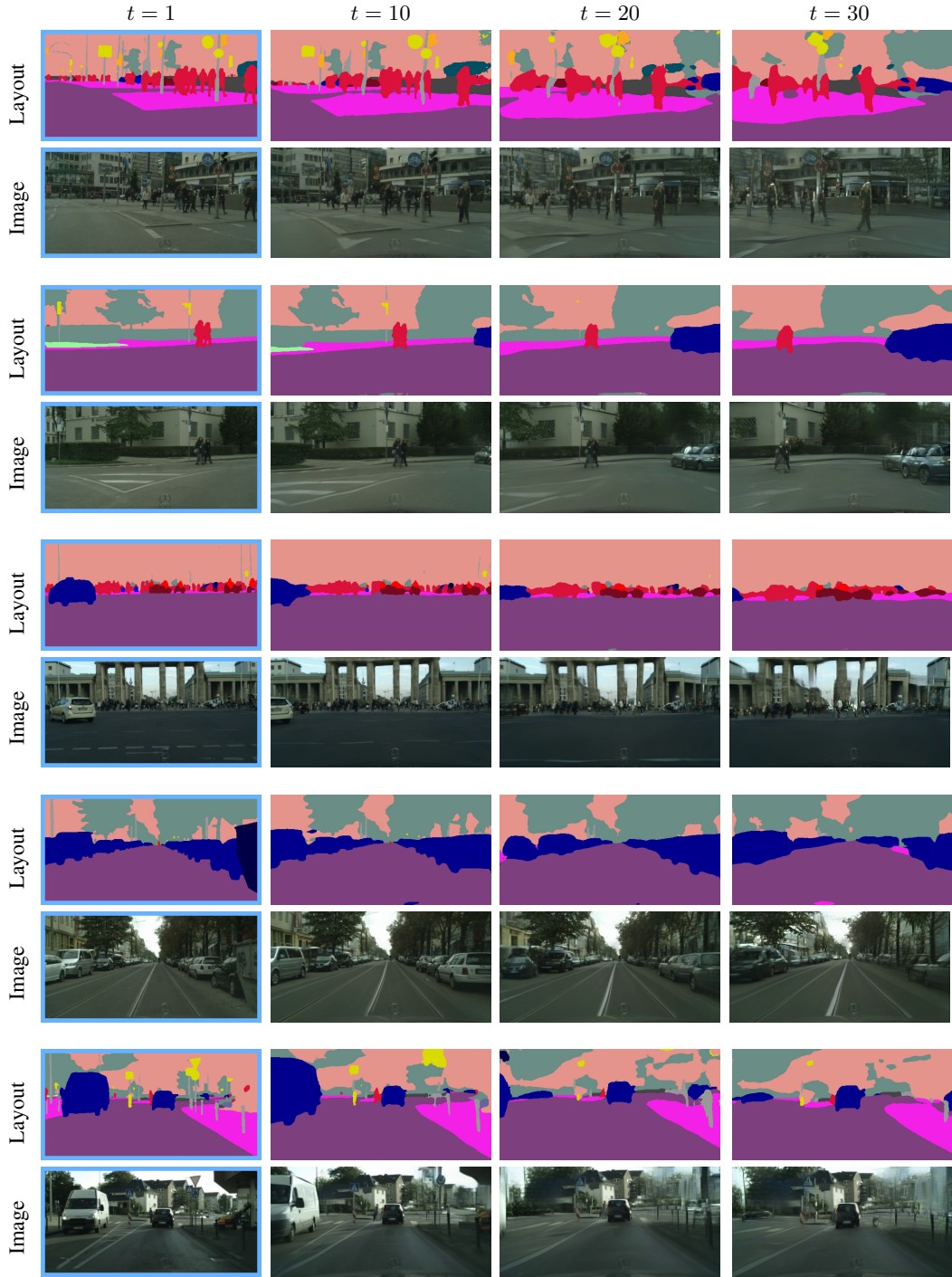

Figure D2: Future video synthesis of 30-frame videos given 3 starting layout-frame pairs on Cityscapes ($256 \times 512$). We show different samples of synthetic layout and image pairs at intermediate timesteps. In this case, layouts are a byproduct of the synthesis process, *i.e.*, they are synthesized alongside predicted frames.

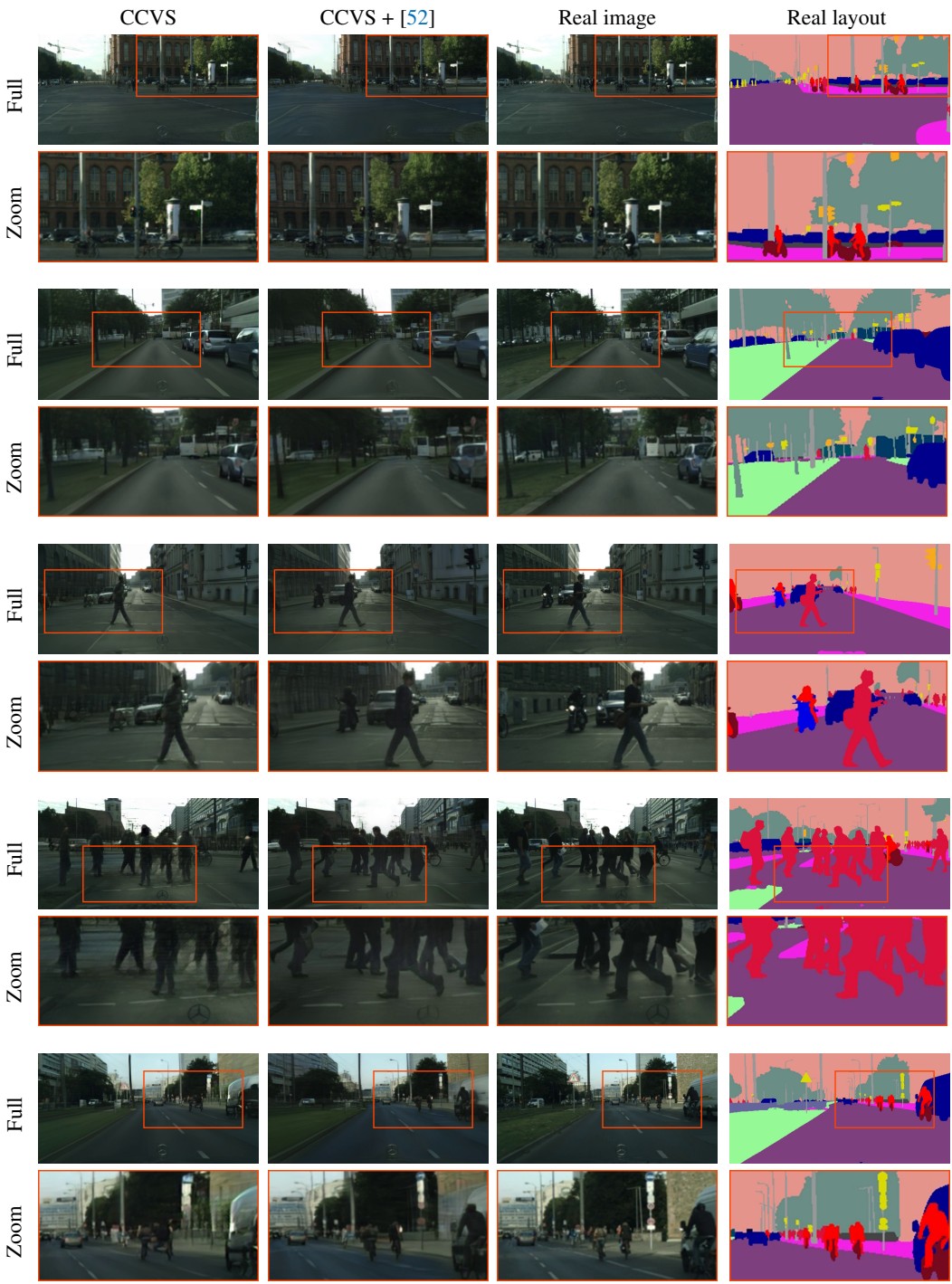

Figure D3: Reconstruction of the 30$^{th}$ frame given the real 1$^{st}$ frame and both compressed features and layouts for subsequent timesteps. We compare our method with and without SPADE decoding blocks [52] to the expected real image and layout at the same 30$^{th}$ timestep. We zoom in specific areas to facilitate comparisons. We observe better alignment and more precise details for objects displaying complex motion when SPADE is used.

Table D3: Sound-conditioned video synthesis on AudioSet-Drums ($64 \times 64$).

| Method | Cond. | Audio | SSIM↑ | | | PSNR↑ | | |
|---|---|---|---|---|---|---|---|---|
| | | | $t = 16$ | $t = 30$ | $t = 45$ | $t = 16$ | $t = 30$ | $t = 45$ |
| CCVS (*ours*) | 1 | | $0.952_{\pm 0.005}$ | $0.942_{\pm 0.009}$ | $0.925_{\pm 0.002}$ | $28.1_{\pm 0.5}$ | $26.8_{\pm 0.5}$ | $25.8_{\pm 0.6}$ |
| CCVS (*ours*) | 1 | ✓ | $\mathbf{0.955}_{\pm 0.006}$ | $\mathbf{0.943}_{\pm 0.008}$ | $\mathbf{0.931}_{\pm 0.010}$ | $\mathbf{28.2}_{\pm 0.5}$ | $\mathbf{27.1}_{\pm 0.4}$ | $\mathbf{26.3}_{\pm 0.4}$ |

Table D4: Point-to-point video synthesis on BAIR ($64 \times 64$). We follow [75] and synthesize 30-frame videos conditioned on the start and end frames. For each of the real test videos we produce synthetic ones (100 predicted samples) and compute the pairwise SSIM between real and synthetic frames. As a quality assessment, we check if the real video is represented among synthetic ones by reporting the best SSIM among samples ("Best"). We estimate the diversity of the predictions by computing the variance of the SSIM across samples with the true video as reference ("Div."). We evaluate the control point consistency by measuring the SSIM between the last synthesized frame and the corresponding real one for all samples ("C.P.C."). Finally, we assess the reconstruction quality for the different methods by using latent features extracted from real frames at intermediate timesteps instead of predicting them ("Rec."). Metrics are computed in a $95\%$ confidence interval like in [75].

| Method | Cond. | SSIM↑ | | | |
|---|---|---|---|---|---|
| | | Best | Div. (1E-3) | C.P.C. | Rec. |
| SVG [15] | 2 | $0.845_{\pm 0.006}$ | $\mathbf{0.716}_{\pm 0.166}$ | $0.775_{\pm 0.008}$ | $0.926_{\pm 0.003}$ |
| SV2P [2] | 2 | $0.841_{\pm 0.010}$ | $0.186_{\pm 0.021}$ | $0.770_{\pm 0.009}$ | $0.847_{\pm 0.004}$ |
| P2PVG [75] | 2 | $0.847_{\pm 0.004}$ | $0.664_{\pm 0.049}$ | $\mathbf{0.824}_{\pm 0.015}$ | $0.907_{\pm 0.006}$ |
| CCVS (*ours*) | 2 | $\mathbf{0.857}_{\pm 0.006}$ | $0.454_{\pm 0.041}$ | $\mathbf{0.824}_{\pm 0.007}$ | $\mathbf{0.960}_{\pm 0.005}$ |

**Point-to-point synthesis.** We explore further the performance of CCVS on the point-to-point synthesis task which we introduced in Section 4 of the article. We show some quantitative results against different baselines in Table D4. CCVS is on par or significantly outperforms the state of the art on 3 out of 4 metrics. It produces more faithful reconstructions when latent features are known. CCVS is more likely to synthesize the true outcome among 100 predicted samples compared to baselines. It has a good control point consistency (*i.e.*, the last synthetic frame is close to the corresponding ground-truth), but lower diversity than prior arts. We note, however, that SSIM diversity may capture the capacity of producing different trajectories between start and end points, but may also increase due to visual quality degradations. Like previous works [75], CCVS is able to produce videos of arbitrary length between the target start and end frames. We recall that the end frame is used as a video-level annotation in the transformer model $\mathcal{T}$ (Figure 2) so that it is taken into account when predicting frames at earlier timesteps. To allow any timestep for the end frame (possibly greater than the capacity of $\mathcal{T}$), we make its positional embedding reflect the gap with current predicted frames. Namely, the positional embedding of the end frame is updated as the sliding window moves forward in time and gets closer to the end timestep. We show qualitative results for different video lengths in Figure D4.

**Class-conditioned synthesis.** Next, we conduct a qualitative study of video synthesis conditioned on a class label on the Weizmann dataset [27], which consists of short clips captured from a fixed viewpoint from 9 subjects, each performing the same 10 actions. We use 8 subjects for training and leave 1 out for testing. CCVS tackles this task by specifying the class label as a video-level annotation in $\mathcal{T}$ (Figure 2). We can make the held-out subject perform the learned actions accurately given a single input image (Table D5). Or we can synthesize new videos without any prior knowledge other than the desired class label (Table D6).

**Qualitative ablation.** We repeat the ablation study of the autoencoder conducted in Table 1 on BAIR [19], by giving some qualitative samples of the reconstruction this time. The reconstruction results of 30-frame videos given the first frame and the compressed features at subsequent timesteps can be found in Table D5. To best compare different ablations of CCVS, we use the same input data for all. A model trained with the $L_1$ loss in RGB space reconstructs the static background and the robotic arm well, while other objects look very blurry. When using the same loss in the feature space of a VGG net [59], objects display higher-frequency details, but look transparent. Having the image discriminator $\Delta_i$ allows much more realistic outcomes, although these are still

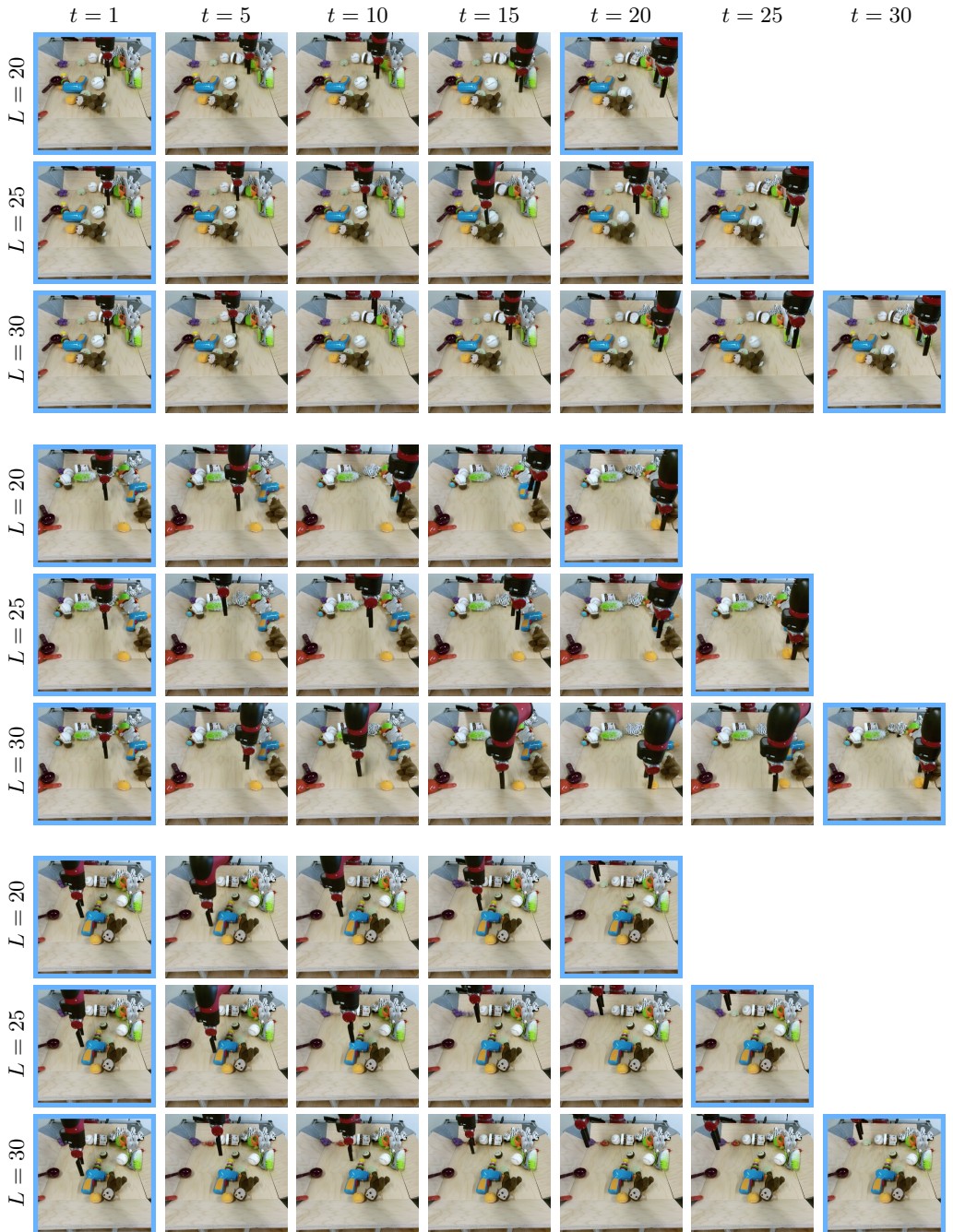

Figure D4: Videos of various lengths ($L \in \{20, 25, 30\}$) synthesized with our point-to-point method given a pair of start and end frames on BAIR ($256 \times 256$). Note that the same model is used to synthesize videos of different lengths. Our approach generalizes to arbitrary lengths between start and end frames by taking the length into account in the positional embedding of the end frame.

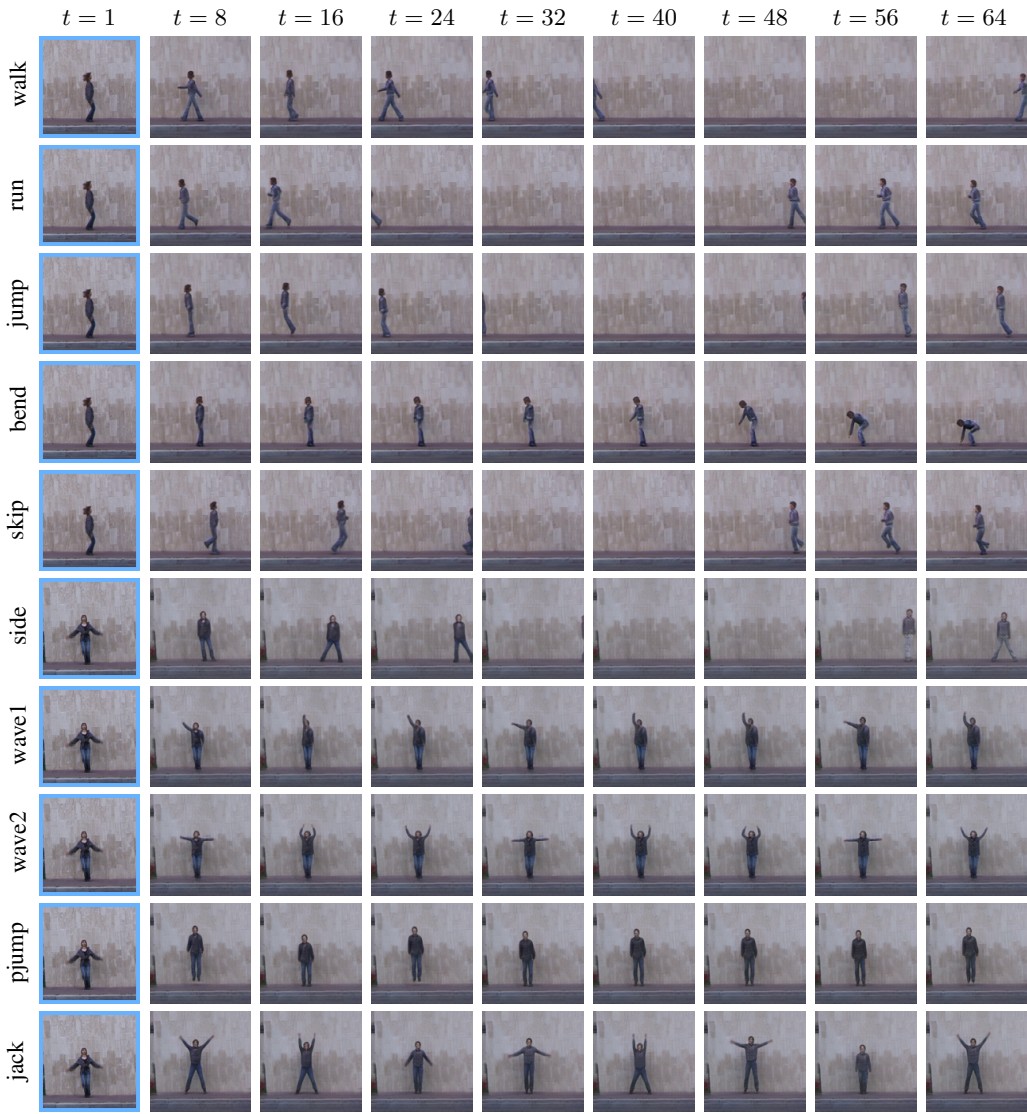

Figure D5: Qualitative samples of class-conditioned synthesis from a single image on Weizmann (128×128). We use one conditioning frame from a held-out subject (not seen during training), that is, a side view for lateral actions ("walk", "run", "jump", "bend" and "skip") and a front view for frontal actions ("side", "wave1", "wave2", "pjump" and "jack").

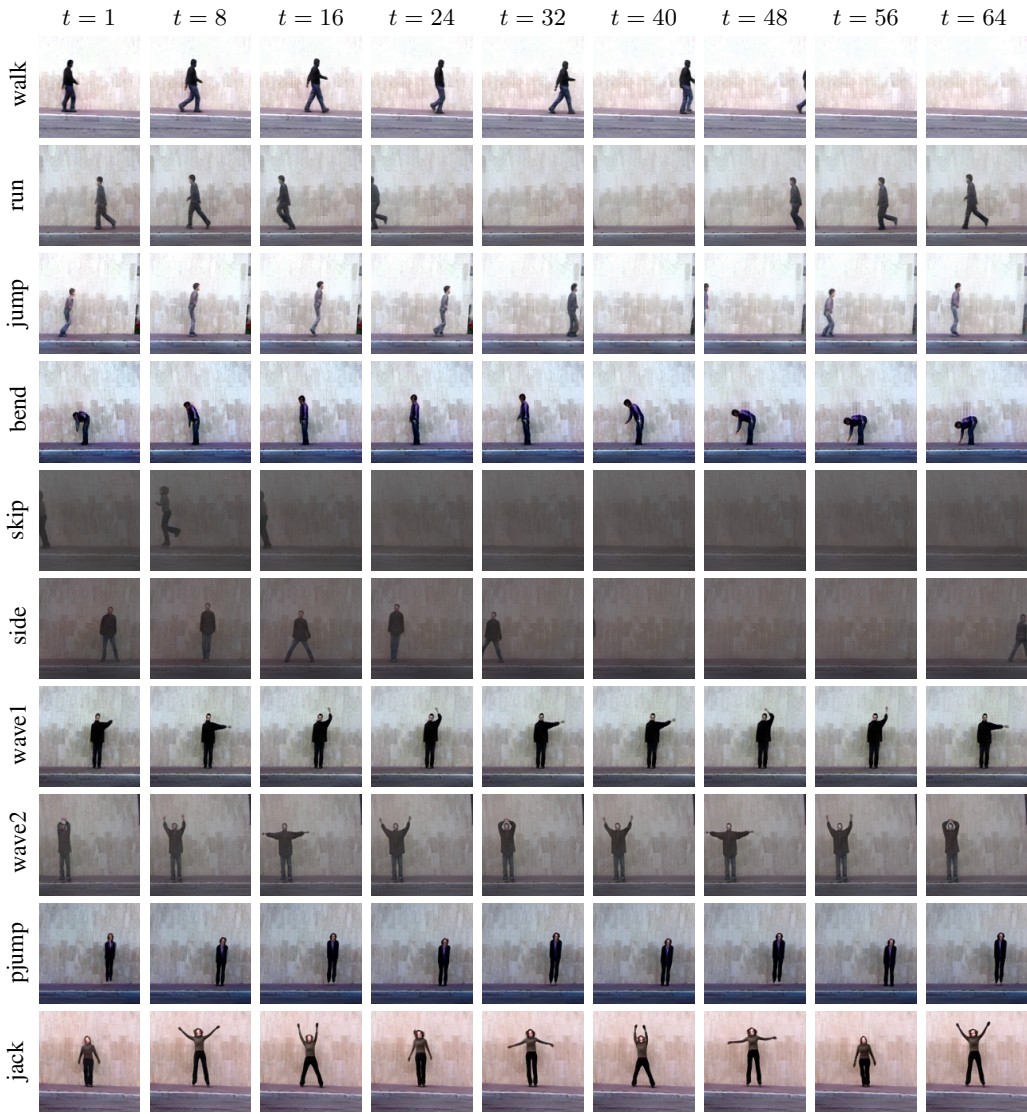

Figure D6: Qualitative samples of class-conditioned synthesis without input image on Weizmann (128×128). Videos are synthesized from the class label alone. The diversity of brightness, contrast and saturation levels reflects the augmentations we use during training. This is necessary to avoid severe overfitting due to the small size of the dataset.

not temporally consistent, nor do they faithfully correspond to the ground-truth video. Having the temporal discriminator $\Delta_t$ seems to improve over the temporal consistency issue. Thanks to the flow module $\mathcal{F}$, reconstructed videos are closer to the ground-truth ones, in particular at early timesteps. However, we see that flow imperfections cause synthesis artefacts which add up as time goes by, and result in quality degradation. The self-supervised technique used to learn $\mathcal{F}$ mitigates this shortcoming. Finally, larger context windows and longer training times help to better recover details (see the more realistic texture and shape of manipulated objects for instance). The final proposal creates high-quality videos almost indistinguishable from real ones, and shows true interactions with objects (a ball and a colorful gun in this example). We see that in case of disocclusion (around $t = 20$ the upper right corner is revealed), our model is able to synthesize plausible arrangements of objects despite lossy compression. These new objects remain consistent as the synthesis process unfolds (see $t = 30$) thanks to the autoregressive nature of our approach.

**Long-term synthesis.** We assess the ability of CCVS to generalize to long-term prediction through a qualitative study. We compare, for a model trained on sequences of length 16, the synthesis of 90-frame videos conditioned on a single frame and using different context windows for the flow module: 0 (not using the flow), 1, 4 and 16. Results on BAIR, Kinetics and AudioSet-Drums are available in Figure D7. When the flow is not used (size 0, *i.e.*, a naïve extension of [20] or a setup close to [53]) videos are quite unstable and we see colors shifting over time. With a single-frame context window (size 1), there is a greater temporal consistency (*e.g.*, the drum sticks are better preserved on AudioSet-Drums, the semantic structure shows higher fidelity to the input frame on BAIR or Kinetics), but in the long-term, synthesis artefacts add up due to the autoregressive decoding which leads to visual deterioration and saturation. These effects stand out the most on BAIR and Kinetics. The proposed multi-frame context extension (Appendix D) reduces this issue (size 4), and the latter becomes barely perceptible with an even larger context window (size 16). We found that even small context windows on AudioSet-Drums are sufficient thanks to the high inter-frame similarity.

# F   Augmentation strategy

As discussed in Section 3.2, the self-supervision process used to train $\mathcal{F}$ reconstructs a static image $x$ from both its latent representation and an augmented view as context by estimating the flow and mask corresponding to that augmentation. We illustrate this process on the Cityscapes dataset in Figure F1. We recall that, in this case, the context image can be written as $x_c = o_c \otimes \mathrm{B}(\mathrm{A}(x))$, where $o_c$ is an occlusion mask, and B and A some blurring and augmentation (*i.e.*, spatial transformation) functions respectively. We can invert the flow corresponding to transformation A thanks to Algorithm C1. The output corresponds to the target "Real Flow" in the illustration. As for the target "Real Mask", it is obtained by warping the occlusion mask with the inverted flow. Looking at synthetic reconstructions, we see that visible details in the context are well recovered despite the augmentation. Indeed, real and synthetic flows match in non occluded areas (indicated by the dashed red line). Moreover, adversarial training produces high-quality images from blurred and occluded contexts. We notice that textures do not always match in real and synthetic images (*e.g.*, building windows, car appearance). Yet, thanks to recovery losses, the semantic structure of the scene is preserved. Finally, we see that darker and lighter areas of the synthetic masks correspond to the same areas in the real ones. The darker the synthetic masks, the more features are updated from context. The *whitish* part indicates that in occluded areas the final image is reconstructed from its latent representation alone, and the *greyish* part shows that for other regions context is effectively reused (and latent representations still contribute to the final outcome). It is interesting to see that edges are visible in the synthetic masks. This indicates that those regions may be easier to detect (darker color) and play an important role in the image alignment process (estimation of the flow).

Table D5: Qualitative ablation study of the autoencoder on BAIR ($256 \times 256$). Reconstruction of 30-frame videos given the first frame and the compressed features at subsequent timesteps.

| Self-recovery | | | | Ctxt.-recovery | | | Reconstructed frames | | | |
|---|---|---|---|---|---|---|---|---|---|---|
| RGB | VGG | $\Delta_i$ | $\Delta_t$ | $\mathcal{F}$ | Sup. | Ctxt. | $t = 2$ | $t = 10$ | $t = 20$ | $t = 30$ |
| ✓ | | | | | | 0 | | | | |
| ✓ | ✓ | | | | | 0 | | | | |
| ✓ | ✓ | ✓ | | | | 0 | | | | |
| ✓ | ✓ | ✓ | ✓ | | | 0 | | | | |
| ✓ | ✓ | ✓ | ✓ | ✓ | | 1 | | | | |
| ✓ | ✓ | ✓ | ✓ | ✓ | ✓ | 1 | | | | |
| ✓ | ✓ | ✓ | ✓ | ✓ | ✓ | 8 | | | | |
| ✓ | ✓ | ✓ | ✓ | ✓ | ✓ | 15 | | | | |
| *Training longer* (num. epochs $\times 3$) | | | | | | | | | | |
| *Oracle* | | | | | | | | | | |

"Sup.": self-supervision of $\mathcal{F}$;    "Ctxt.": number of context frames taken into account (in $\mathcal{F}$) when decoding current frame (in $\mathcal{D}$).

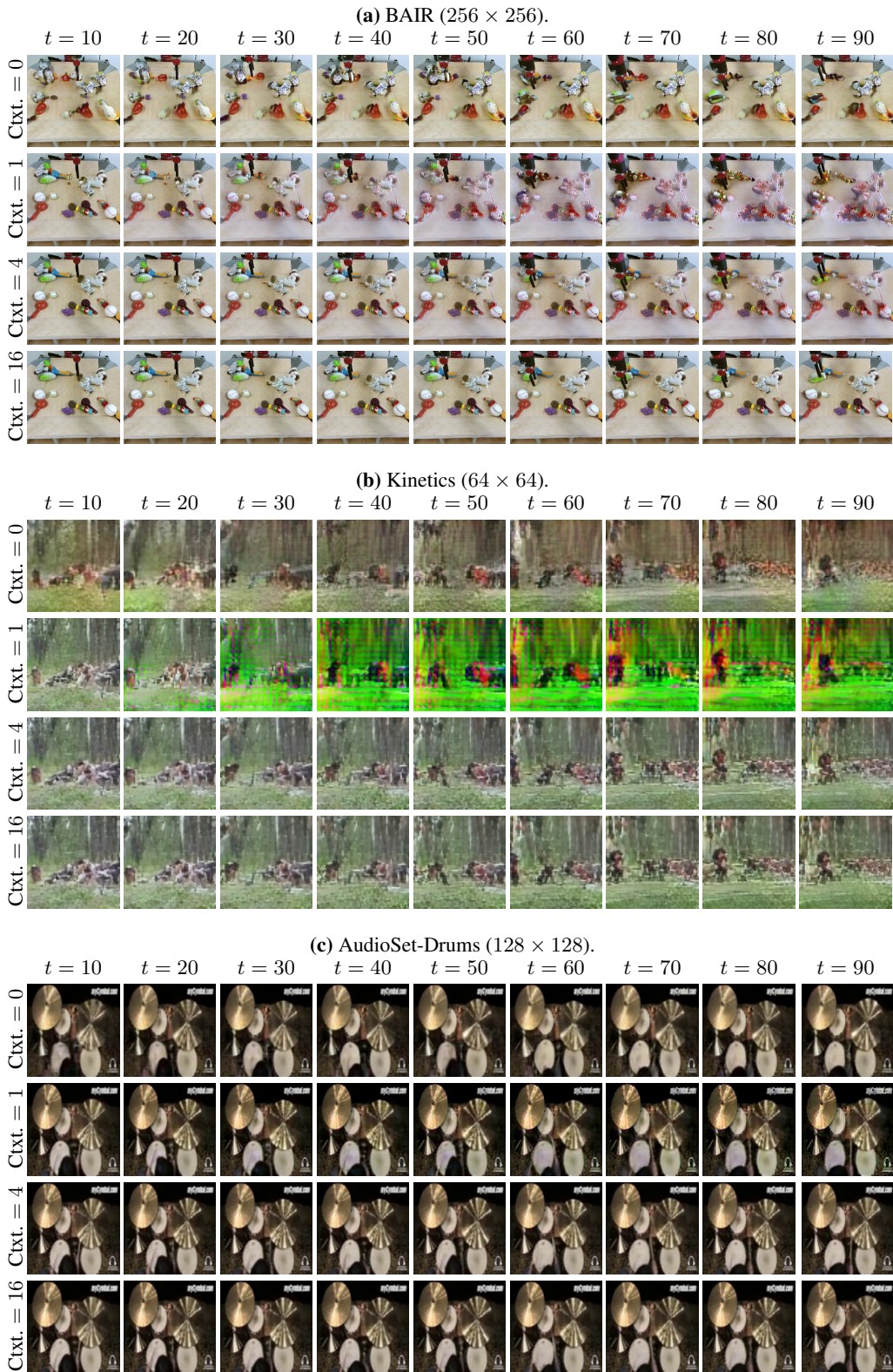

Figure D7: Qualitative samples for long-term generalization of a model trained on 16-frame videos to the synthesis of 90-frame ones conditioned on one frame. We compare different context windows ("Ctx.") for the flow module: 0 (not using the flow), 1, 4 and 16. Zoom in for details.

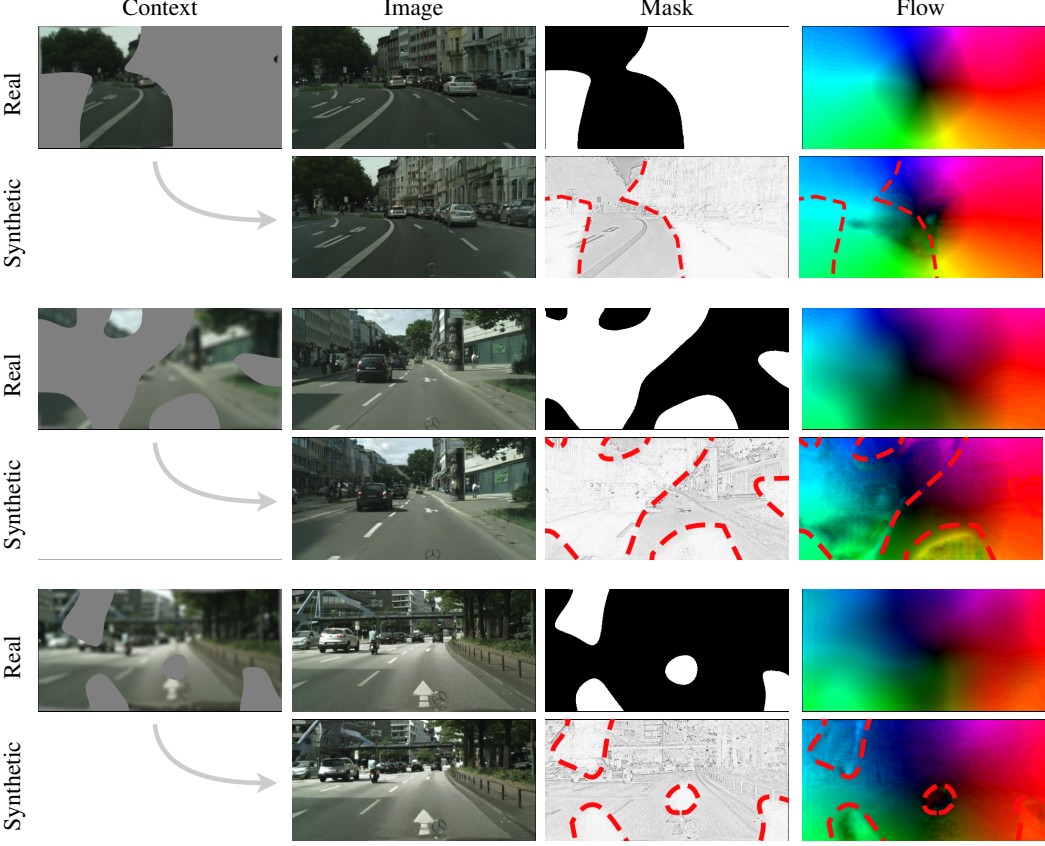

Figure F1: Some examples of reconstructed image, fusion mask and optical flow from self-augmented views on Cityscapes ($256 \times 512$). For better visualization, we use a dashed red line in the synthetic mask and flow to indicate the outline of the real mask. Training losses encourage real and synthetic masks to be close within occluded context areas (white region), and real and synthetic flows to be close within the visible context areas (black region).