# OpenReview forum: "CCVS: Context-aware Controllable Video Synthesis"
_NeurIPS.cc/2021/Conference — NeurIPS 2021 Poster_

### Official Review · Reviewer_ZPQv · 2021-07-16

**Rating:** 6
**Confidence:** 3

**Summary:**

This paper introduced a conditional video prediction algorithm based on VQ-VAE [57] and Vision Transformer (ViT). Similar to VQ-VAE, the frame-level discrete representation is learned by autoencoder with vector quantization, whose (conditional) prior on the latent representation is modeled by the variants of ViT. The authors also incorporated a useful inductive bias for video prediction by warping features based on optical flow estimation, as in [63]. The practical impacts of the proposed method are demonstrated in various short-term video prediction tasks, showing competitive performance to the previous SOTAs.

**Limitations And Societal Impact:**

Yes.

**Main Review:**

Pros:
- This paper proposes a general video prediction framework that can incorporate various structured conditional signals. The experiment results show that indeed the generated results are reasonable across diverse conditioning signals.
- The performance of the proposed method seems to be superior/competitive to the previous SOTAs on video prediction. Thorough ablation studies were helpful to understand the contribution of the model.

Cons:
- The proposed method cleverly combined many “working ideas” from the previous methods. Although its motivation is sound and performance is decent, it (1) lacks novel technical contributions and (2) often misses proper citations. In terms of technical novelty, it follows the idea of combining VQ-VAE with the Transformer as in previous works (e.g., [46]). Although the authors improved its architecture by modifying both the Transformer and VQ-VAE autoencoders, the modifications are also largely based on the existing techniques (ViT [A1] for Transformer and feature warping [63] for autoencoder). In terms of citation, the paper misses many important citations, making readers unfamiliar with this domain difficult to assess its contribution. For instance, the paper misses citation to ViT [A1] although it largely borrows their architecture. The feature warping in Eq.(4) was also introduced without proper citation to its original paper ([63];  Eq.(4)).

[A1] Dosovitskiy et al., An Image is Worth 16x16 Words: Transformers for Image Recognition at Scale, In ICLR, 2021

- In principle, the proposed method can generate diverse futures. It would be great if authors can provide some qualitative results of diverse synthesis results (images or videos) conditioned on the same input. The current supplementary file provides results only on different conditionings.

- It would be interesting to see if the proposed ViT+VQ-VAE architecture has a strength in long-term generalization in testing (e.g., learned to generate 15 frames into the future and evaluated to generate 50 frames in testing). Some experiments/comparisons to the other methods would be helpful to get more insights.

- The current draft misses important implementation details in the main paper; for instance, it was really unclear to me how the flow estimator is trained (Ln 189-199).

===== Post-rebuttal update ======

The authors addressed some of my major concerns in the rebuttal. I think the paper misses proper citations/discussions to the prior works (even if some of them are rather "conventional"). Although the technical contribution of this paper seems to be not significant, the results are promising. Overall, it seems to me that it is reasonable submission to NeurIPS. I mildly recommend the acceptance.


**Time Spent Reviewing:**

3

---

> ### Author Response · Authors · 2021-08-10
> **Detailed Response to Reviewer ZPQv**
>
> **Missing references and novelty of the approach.** First, we thank the reviewer for pointing out [A1] which is indeed a recent and significant success in adapting transformers [58] to process visual data, and whose architecture is closely related to ours. In particular, their discussion on the design of positional embeddings to process image data was helpful to us when designing embeddings suitable for video data. Our architecture differs from theirs in that we use a causal mask in the self-attention operations to enable autoregressive sampling like in the language model [3].
>
> Second, we believe that the update rule mentioned by the reviewer (Eq. (1) of the submission) is rather classical as a mean for warping and fusing two streams of spatial information (feature maps or images): Eq. (4) of [63], but also, Sec. 3.3 of [23] or Eq. (3) of [Semantic Video CNNs through Representation Warping, ICCV 2017] (as pointed out by reviewer Q5Cg). We will clarify this before introducing Eq. (1). Moreover we have successfully extended this mechanism to multi-frame context (see Sec. C of the supplementary material), with meaningful qualitative and quantitative performance gains, while not having to learn any additional parameter. To the best of our knowledge, this has not been done before.
>
> **Visual evidence of diversity.** The qualitative study requested by the reviewer corresponds exactly to the quantitative study suggested by reviewer mX9P. For the three datasets (Kinetics, BAIR and AudioSet-Drums), we show 100 mosaïcs of 9 videos conditioned on the same input frame. Each video amounts to 15 frames including the conditioning one. Videos are available here [ https://drive.google.com/drive/folders/1F4jW-kPyf1rEIDuenZqvgxbH_Vf3VG3u?usp=sharing ]. Although they share the same starting point, synthesized videos show various types of motion. We will include these qualitative samples with detailed comments alongside the quantitative analysis on diversity proposed by reviewer mX9P.
>
> **Generalization to long-term prediction.** The AudioSet-Drums videos in the supplementary material (see ''ccvs_video.mp4'' file) already demonstrate prediction for a model trained on sequences of 16 frames and used to produce a video totalling 90 frames. As suggested by the reviewer, we have conducted for this rebuttal an additional qualitative study of long-term synthesis. We show, for a model trained on sequences of length 16, 90-frame videos conditioned on a single frame and using different context windows for the flow module: 0 (not using the flow), 1, 4 and 16. This study is conducted on BAIR, Kinetics and AudioSet-Drums and videos are available here [ https://drive.google.com/drive/folders/17uLies17nCifkfns0gKdxk7XTD490g1A?usp=sharing ]. For a given sequence of predicted features, corresponding videos are shown side-by-side with an increasing context window from the left to the right. When the flow is not used (size 0, i.e., a naïve extension of [17] or a setup close to [46]) videos are quite unstable and we see colors shifting over time. With a single-frame context window (size 1), there is a greater temporal consistency, but in the long-term, synthesis artefacts add up due to the autoregressive decoding which leads to visual deterioration and saturation. These effects stand out the most on BAIR and Kinetics. The proposed multi-frame context extension reduces this issue (size 4), and the latter becomes barely perceptible with an even larger context window (size 16).
>
> **Training of the flow module.** We will follow reviewer Q5Cg's suggestion of reorganizing the paper to better convey the inner mechanisms of the flow module, *i.e.*, use the additional page to move some explanations from the supplementary material to the main paper. As explained in the response to reviewer mX9P, we will also clarify the way this module is trained. We will mention that Eqs. (4) and (6) are used to update the module (on top of Eq. (7)) as it is trained jointly with the autoencoder. We will also explain it in the text so that it is explicit and easy to understand. Finally, we note that the code containing the concrete implementation of the different modules was included in the supplementary material and that we plan to release it publicly by making it open source.

---

> > ### Comment · Reviewer_ZPQv · 2021-08-21
> > **Response to authors' rebuttal**
> >
> > I appreciate the authors' response. The rebuttal addressed some of my major concerns on experiments. In revision, please properly acknowledge the closely related works, especially ViT [A1], and include sufficient discussions on implementation details and experiments on Kinetics and AudioSet-Drum datasets. Also, please add results discussed in the rebuttal period to the paper.

---

> > > ### Author Response · Authors · 2021-08-23
> > > **RE: Response to authors' rebuttal**
> > >
> > > We agree with the reviewer's suggestions and commit to following them if the paper is accepted.

---

### Official Review · Reviewer_mX9P · 2021-07-16

**Rating:** 6
**Confidence:** 5

**Summary:**

This paper proposes a method for video synthesis that combines an encoder / decoder architecture that computes discrete representation of images with a transformer network that predicts those discrete representations into the future. The encoder and decoder are connected through a novel optical flow module that warps encoder features into the decoder in order to predict how objects will move. In experiments, the authors perform thorough ablations of the proposed method, and show that they outperform most baselines.

**Limitations And Societal Impact:**

They have discussed potential negative societal impact, but as far as I can tell there is no discussion of limitations.

**Main Review:**

Strengths:
+ New feature level warping strategy for future video synthesis
+ Thorough ablations
+ Outperforms most baselines

Weaknesses:

- Flow module and latent variables:
In Figure 1, we can see that there are two sources of information that contribute to generating frame X_s. First source is Z_s and the second source is F. From observing the provided videos, it looks like the synthesized frames are mostly a result of warping which means that Z_s is potentially being completely ignored, and thus, there is no point in having it to begin with since F provides all necessary information. The authors claim that FVD should provide a measure of diversity since we are comparing the generative distribution and real distribution. However, since the entire dataset is used to compute this number, and not individual videos, then it is hard to claim diversity unless the generation scenario is unconditional (i.e., no encoder, so F becomes impossible to apply). Is it possible to provide diversity metrics for the conditioned scenarios? For example, sample multiple trajectories from the same condition and measure the distance between them. It would be good to provide this for all datasets, and not just the BAIR dataset, to make sure this applies to all experiments.


- How is the flow module learned?
It is not clear to me how the optical flow module is learned. Is this completely learned using the frame reconstruction signal? So, features from the encoder are warped into the decoder using the optical flow module and the reconstruction error signal flows through this operation?


- Visual comparisons against baselines (or best baseline)?
I was not able to find any visual comparisons between the proposed method and baselines. This would be good to have to see if/how the baselines fail where the proposed method succeeds.


- Thin experimental section except for BAIR experiments:
The experimental section presents a good amount of content for the BAIR experiments. However, the subsections for Kinetics and AudioSet-Drum are very thin. There are no real discussions or analysis of what was observed in these experiments. I understand that the page limits are usually not ideal, but this is where authors have to figure out how to organize the paper in a way that sections do not look too thin.


Conclusion:
The proposed method is able to generate good looking videos by combining discrete variables, warping, transformers and Adversarial objectives. Having said that, there are a few questions that I have about the method to get a good idea about the performance of the proposed method. My current score is slightly above borderline, but I am willing to increase my score depending on the author’s response.


**Time Spent Reviewing:**

4

---

> ### Author Response · Authors · 2021-08-10
> **Detailed Response to Reviewer mX9P**
>
> **Necessity of encoded features.** The scenario described by the reviewer where Z_s is ''potentially being completely ignored'' would apply if a frame was fully determined from the preceding one. This is not the case due to the inherent stochastic nature of future prediction in the considered datasets. We have conducted a simple experiment to demonstrate that features Z_s are actually used and that they actively drive the dynamics of the scene: We have repeated the ''reconstruction'' experiment from Table 1 for which we compare the final model (last line) with an ablation with fixed Z_s (*i.e.*, taking Z_i=Z_1 for all timesteps i) and obtained the following results:
>
> |          Z_s |  FVD ↓  | PSNR ↑ |
> |-------------|------------|-----------|
> |     Fixed   |  556±12 |    18.9    |
> | Dynamic |    45±1   |    26.8    |
>
> We note that the reconstruction performance is much poorer when we force Z_s to remain constant for all timesteps. Qualitatively, looking at the synthesized videos, we see that they do not exhibit any motion, as one would expect when fixing the encoded features.
>
> Moreover, thanks to the fusion masks estimated by the flow module F and used in Eq. (1), it is possible to observe how much information comes from Z_s and from F respectively when generating X_s. Figure E1 of the supplementary material shows some examples of estimated masks on Cityscapes. Those are white when the source is Z_s, and black when it is F. In practice, we see that they are white when the context is occluded and mostly grey in other regions. Thus, final videos are as much the result of context warping than direct decoding in non-occluded regions for this dataset.
>
> **Diversity metric for conditioned scenarios.** We agree with the reviewer that our claim that FVD reflects diversity should be reformulated to encompass the sole unconditional case. The stronger the conditioning signal, the less the FVD metric reflects diversity, and another metric is indeed necessary to cover the conditioned scenarios. We have followed the reviewer’s suggestion to compare the diversity of trajectories for a given conditioning input: We have taken a 29-frame video, used the 15th frame as conditioning input to sample 10 synthetic trajectories, and examined the 15→29 and 15→1 sequences as real trajectories. We have then computed the mean pairwise similarity between synthetic trajectories, and used the real ones to approximate the ground-truth inter-trajectory similarity. It is important to have a point of comparison by estimating the real statistics since the degree and variance of scene transformations over time can widely differ from one dataset to another. Although the 15→1 sequence is reversed and may result in unnatural motion, it allows to constitute pairs of real trajectories starting from the same frame (which otherwise would not be possible). We have repeated this experiment 100 times on BAIR, Kinetics and AudioSet-Drums, and report below the similarity among real and synthetic trajectories in terms of PSNR and SSIM, the idea being that the lower the similarity among trajectories, the higher the diversity is.
>
> `BAIR`
>
> |                 |     PSNR     |       SSIM      |
> |-------------|---------------|----------------|
> | Synthetic | 20.92±0.54 | 0.869±0.015 |
> |     Real     | 19.97±0.49 | 0.852±0.014 |
>
> `Kinetics`
>
> |                 |     PSNR     |       SSIM      |
> |-------------|---------------|----------------|
> | Synthetic | 22.28±0.77 | 0.658±0.023 |
> |     Real     | 22.62±0.92 | 0.646±0.034 |
>
> `AudioSet-Drums`
>
> |                 |     PSNR     |       SSIM      |
> |-------------|---------------|----------------|
> | Synthetic | 29.95±0.58 | 0.962±0.003 |
> |     Real     | 27.75±0.74 | 0.938±0.007 |
>
> Looking at individual datasets, the real trajectories seem slightly more dissimilar (or diverse) than the synthetic ones. This may be due to using reversed sequence 15→1 which by construction shows reverse motion (at least in the short term) compared to 15→29. Still, similarity scores for synthetic and real are consistent when considering all datasets, which means that the overall diversity of videos generated from a given conditioning input complies with the estimated real statistics. We invite the reviewer to look at the qualitative samples requested by reviewer ZPQv under the same experimental setup which are available here [ https://drive.google.com/drive/folders/1F4jW-kPyf1rEIDuenZqvgxbH_Vf3VG3u?usp=sharing ].
>
> **Training of the flow module.** The flow module is trained jointly with the autoencoder. Its parameters are therefore updated by gradients which arise from the same reconstruction error signals as the decoder (Eqs. (4) and (6)) on top of the contextual loss (Eq. (7)) for self-supervision of the mask and flow predictions. The reviewer is correct in saying that the flow module lets gradients flow from the decoder to the encoder through the warping operations. In addition, we note that in the decoding of a frame for a given timestep, encoder features which are used in the flow module always correspond to different timesteps to avoid trivial solutions. We realized that mentions of the flow module were missing in Eqs. (4) and (6). We will fix this and add further explanations in our revision.
>
> **Qualitative comparisons to baselines.** In Table D4 of the supplementary material we propose a qualitative ablation study for our method. In the corresponding text we discuss the contribution of each of our framework’s components to the overall visual quality and fidelity. In particular, one can assess the effect of the flow module (and the proposed multi-frame extension), a major technical distinction and improvement over prior work [46]. For more visual comparisons to prior arts see the detailed response to reviewer Q5Cg.
>
> **Organization of the experimental section.** We will follow the reviewer’s suggestion and rebalance the space allocated to each experiment subsection in our revision. We will further analyze Kinetics and AudioSet-Drums results in the main paper. Moreover, we intend to include the qualitative study on AudioSet-Drum proposed by reviewer Q5Cg in the main paper, if space permits.
>
> **Limitations.** Major limitations are highlighted in the paper’s checklist. We will make sure that they are more visible directly in the main body of the paper.

---

> > ### Comment · Reviewer_mX9P · 2021-08-28
> > **Response to author's response**
> >
> > I would like to thank the authors for their response. The authors addressed all of my concerns, and so, I will be increasing my score. I have one last comment regarding diversity. I can clearly see the video diversity in the videos the authors provided (this is a really good save), but the metrics provided do not highlight the diversity. I suggest the authors simply measure the pixel-wise distance between the predicted videos similar to this paper: https://arxiv.org/pdf/1901.09024.pdf (Table 5). If the distance is not close to zero, we can conclude there is diversity in the generated data. Given that datasets such as BAIR Robotpush have a large portion of static background, the authors could also measure the pixel distance between the moving parts of the video similar to how this work measures performance of the moving parts: https://arxiv.org/pdf/1706.08033.pdf (Supplementary material Section C).

---

> > > ### Author Response · Authors · 2021-08-31
> > > **RE: Response to author's response**
> > >
> > > **Diversity metric for conditioned scenarios.** We thank the reviewer with his/her comments. First, the PSNR metric as used in our experiment is in fact a measure of pixel-wise similarity, so pixel-wise distance increases as PSNR decreases. Moreover, we can also conclude there is diversity if the SSIM is not close to one. Second, we have followed the reviewer's suggestion and actually measured the mean pixel-wise distance among synthetic trajectories for the three datasets:
> > >
> > > |  | BAIR | Kinetics | AudioSet-Drums |
> > > |---|---|---|---|
> > > | Whole frames | 19.53±2.6 | 20.85±2.9 | 2.25±0.26 |
> > > | Moving parts only | 136.88±12.27 | 48.16±13.83 | 65.45±6.00 |
> > >
> > > The pixel-wise distances are obtained by applying a $1.0 \times 10^{−3}$ factor (omitted in the table for clarity). The results on whole frames for BAIR are coherent with the ones given in Table 5 of *Yang et al.* ( https://arxiv.org/pdf/1901.09024.pdf ). We obtain similar results on Kinetics. On AudioSet-Drums the pixel-wise distance on whole frames is lower by a factor 10, like the diversity measurement obtained by *Yang et al.* on the KTH dataset. This is explained by the fact that, in both AudioSet-Drums and KTH, motion is quite repetitive and involves a limited portion of the frame. We also report results featuring only the moving parts of the scene, as proposed in *Villegas et al.* ( https://arxiv.org/pdf/1706.08033.pdf ): We compute the flow between consecutive frames and mask non-moving regions where the magnitude of the flow is less than 20% of the maximum magnitude. This results in an increased pixel-wise distance on the three dataset. This metric better highlights the diversity when the static parts cover a large portion of the scene. We will include these additional observations in our revision.

---

### Official Review · Reviewer_Q5Cg · 2021-07-17

**Rating:** 6
**Confidence:** 3

**Summary:**

This submission presents a conditional video synthesis model. This work is largely based on the recent progress of the Transformer-based autoregressive model [17] but with video extension. Differences over [17] are 1) learning-based flow module for temporal consistency, 2) conditioning for controllability, and 3) temporal embedding.

Overall, the submission contains a few technical novelties while being questionable whether significant, and the organization could be further improved.
Other than those, the results seem reasonable, and the analyses are well conducted.


=============================================

<After the rebuttal and discussion phase>

The authors' responses have addressed most of this reviewer's concerns, which increases my rating.
The authors are strongly required to reflect all the rebuttal responses in the paper; especially missing reference, acknowledging the conventional works, the additional results, and the preliminary results with the SPADE version.
More discussion about the flow module is recommended.


**Limitations And Societal Impact:**

* The authors argued a positive effect on energy consumption by faster training thanks to the compact code.
* Misuse case discussion is included, and the authors argued that the positive applications outweigh the potential ethical concerns.

The discussions sound reasonable.

**Main Review:**


Pros
- Video extension of [17] introducing flow module, controllability, and temporal embedding
- Simple and general conditioning method using Transformer (frame, sound and pose conditioning)
- Sufficient experiments and study. Nice ablation
- Good quantitative results

Cons
- Questionable effectiveness of the flow module in temporal consistency
- Paper organization and writing could be improved (the use of terminologies are confusing, and notations are abused)
- Hard to feel improvement over the state-of-the-art methods in qualitative results.


<Detail comments>

* Sound-conditioned video synthesis results: Multi-modal controllability is very interesting in generative modeling.
It would be more interesting to show the diverse synthesis results obtained from the fixed visual condition and different sound contexts (e.g., different drum playing sounds). This would allow to clearly understand behaviors and insights of the proposed method w.r.t. conditioning.

* Layout-conditioned video synthesis results: Compared to [40], the results obtained by the proposed method suffer from wobble effects and unstable temporal consistency.
Then, it is questionable what is the benefit of using the flow module.
This is concerning in that the flow-module is one of the main technical contributions and the most technical distinctive extension over [17], which seems to be neutralized by failing to show its effectiveness.

* BAIR results look interesting. The proposed method seems to understand physical interaction well to some extent.

* Missing reference: Flow-based feature warping modules have been used for temporal consistency, e.g., [Semantic Video CNNs through Representation Warping, ICCV 2017]. However, in Line 154, the references are missing, and the discussion about the history is misleading.


<Writing issues>

* Terminologies are confusing: The authors interchangeably used {encoding, embedding, feature, code}, {annotation, condition, state}, etc. Those badly affect the readability of the submission.

* Flow module: By reading about the proposed flow module in the main paper, this reviewer failed to understand how the flow module is operated clearly, and found that the description in the supplementary material is much more specific and clear to understand.
The writing should be reorganized to convey the clear elaboration of the module.

* Fig. 3: The y-axis scale annotation is confusing (log_10). Is it 10 FPS or exp(10) FPS?

* Notation abuse: Dimensions defined in the submission are confusingly described. The authors seem to intend this to appear to be general, but it disturbs understanding about the dimensions. Also, it leads to duplicated descriptions about dimensions in Sec. 3.1
and Sec. 3.2, which can be much condensed. It would be more informative to bring some of the supplementary material contexts by reorganizing the descriptions to make more space.




* Question: Why should the flow inversion be approximated? This reviewer cannot see any reason to use this approximation rather than directly learning to extract backward flow.




**Time Spent Reviewing:**

5

---

> ### Author Response · Authors · 2021-08-10
> **Detailed Response to Reviewer Q5Cg**
>
> **Qualitative comparison to state-of-the-art methods.** We believe that it is difficult to perform objective qualitative comparisons between video synthesis methods and our assessment below is of course subjective. Following reviewer Q5Cg’s request, we provide here links to some qualitative results obtained using state-of-the-art methods, namely [39, 42, 67], and available on their web sites, and kindly invite the reviewer to visually compare to videos obtained by our methods with the same input data and judge for him/herself.
>
> Quantitatively, the state of the art for 64x64 BAIR is [67] (our method ranks second). The authors provide 16-frame synthetic continuations conditioned on the first frame of each of the 256 test sequences at: [ https://drive.google.com/file/d/1Gg7Rj5r0KZUWGRazmQqFy8aZVA1Blhg4/view ]. Samples obtained using our method are available at: [ https://drive.google.com/drive/folders/1ZfwekRgxMgBCv5X_5y__-oM6NwzEWfkD?usp=sharing ]. In our opinion, both [67] and our method produce convincing results: synthetic videos are difficult to tell apart from the real ones on this dataset and at this resolution level.
>
> On more diverse datasets like Kinetics, we significantly outperform [67] from a quantitative viewpoint. The author provide examples of 16-frame continuations conditioned on 5 real frames for Kinetics at [ https://drive.google.com/file/d/1Uy1XMEt-FpM47e1G4O0rfZ0Qsq_e3oh0/view ], and one can see that on several occasions the videos end up freezing. Random samples from our method under the same experimental setup do not have this issue, see [ https://drive.google.com/drive/folders/1ZfwekRgxMgBCv5X_5y__-oM6NwzEWfkD?usp=sharing ].
>
> Quantitatively, the state of the art method for the Kinetics benchmark is [39], whose authors provide continuations for models trained on Kinetics at this link: [ https://drive.google.com/drive/folders/1fvmEm3gOWprWy2IuMFe4DuwEPahe9MwC ]. These synthetic samples are conditioned on the UCF101 dataset as a choice motivated by ethical concerns by the authors while we followed the evaluation protocol of [67]. Although results are not strictly comparable, the two datasets display similar features which should be sufficient for visual assessments. Qualitatively, it seems to us that the videos produced by [39] and our method are roughly of the same visual quality, neither of them exhibiting specific artefacts or visual defects.
>
> In addition, unlike [39, 67] and most prior works (apart from strongly-conditioned ones), our method can be used to produce high-resolution videos thanks to its computational efficiency. The recent PVG approach of [42] is also able to tackle high-resolutions videos. Examples of videos synthesized from 256x256 BAIR using PVG are provided by its authors at  [ https://willi-menapace.github.io/playable-video-generation-website/main.html ]. Videos directly comparable to these (same data and setting) are available in the supplementary material of our submission (see ''ccvs_video.mp4'' file). We invite the reviewer to compare these videos and make his/her own mind, but we believe that the PVG videos sometimes feature synthesized objects with faded colors and faint edges. Likewise, the robot arm sometimes makes objects fade away as it passes over them, instead of truly interacting with them. We have not encountered these issues in our own experiments. We will discuss them (not meant as criticisms of previous work of course) in the revision of our paper if it is accepted.
>
> **Qualitative assessment for sound-conditioned video synthesis.** We have followed the reviewer's suggestion and evaluated the diversity of 90-frame synthetic samples for a shared visual conditioning (one frame) and various random sound contexts on AudioSet-Drums, see [ https://drive.google.com/drive/folders/14188dObEuPu3EJZ2zIHbbR_aL_SDsCsa?usp=sharing ]. Side-by-side videos show that with different audio track inputs, diverse synthetic videos are produced. The drummer hits different regions of the instrument depending on the sound and the matching between visual and audio features seems reasonable. We invite the reviewer to look at the ablation with/without sound that we conducted on the same dataset (Table D2 of the supplementary material) which gives quantitative evidence that the sound context actively drives the synthesis process. We note that the scores are close due to the high inter-frame similarity on this dataset.
>
> **Effectiveness of the flow module.** We propose a qualitative ablation of the flow module to show its effectiveness. Synthetic samples on long-term prediction conditioned on a single frame, available at this link [ https://drive.google.com/drive/folders/17uLies17nCifkfns0gKdxk7XTD490g1A?usp=sharing ] show that, compared to decoding frames in a temporally independent fashion like a naïve extension of [17] or in [46] (leftmost video), using the flow (second video from the left) and especially our multi-frame extension (next video to the right and rightmost video corresponding to context windows of 4 and 16 respectively) improve temporal consistency and visual quality. These observations are in line with the quantitative and qualitative ablation studies in the submission and supplementary material respectively (Table 1 and D4).
>
> **Layout-conditioned synthesis.** For the special case of layout-conditioned synthesis, Figure 1 and 5 in [40] show how a 3D world model can improve consistency of texture and object appearance over time. Our flow module plays a similar role without requiring the construction of such a model. Our simple multi-frame context extension allows us to tackle long-term consistency as shown above in our answer for the ''effectiveness of the flow module''. As opposed to [40], our approach is not limited to static objects since the flow module handles not only camera motion but also object motion. The layout guidance in our framework can easily be strengthened so as to avoid wobble, by replacing ResNet blocks [24] by SPADE blocks [Semantic Image Synthesis with Spatially-Adaptive Normalization, CVPR 2019] in our decoder architecture. This is a common architectural choice in layout-conditioned synthesis (also used in [40]) which we did not include in our original submission for the simplicity of having the same decoder across different tasks.
>
> Following the reviewer’s request, and in accordance with the NeurIPS guidelines, we have conducted preliminary experiments with SPADE blocks. Side-by-side reconstructions for ResNet (left) and SPADE decoding blocks (right) are available here [ https://drive.google.com/drive/folders/1iCovop_upnn84eYmf94gQCa1TtbVfGzY?usp=sharing ]. These reconstructions are obtained by encoding and decoding the real videos, and by using only the first frame as initial context for the flow module. Ground-truth layouts are also fed to the decoder when using SPADE blocks. Preliminary results are encouraging: The combination of SPADE with our flow module handles complex motions better (e.g., humans walking in video n°0, 7, 16) and reduces wobble (e.g., vehicles in video n° 4, 8, 12, riders in video n°11, 12). We plan to extend this analysis on synthetic samples (with temporal prediction of encoded features) and measure the improvement of SPADE blocks for temporal consistency in terms of SSIM and PSNR (in the same spirit as Table D2 of the supplementary material). We could not include this extension here due to lack of time, but it will be part of our revision if the paper is accepted.
>
> **Missing references.** We apologize for this omission. We will adapt the corresponding text and also include [Learning Flow-based Feature Warping for Face Frontalization with Illumination Inconsistent Supervision, ECCV 2020], which uses flow-based feature warping for consistency across different views of the same scene, and other relevant references.
>
> **Writing issues.** We thank the reviewer for his/her suggestions for improving the overall clarity of the paper. We will follow them in our revision.
>
> **Flow inversion approximation.** The reviewer raises an interesting question. With the notation of Sec. 3.2,  swapping the roles of d_s and e_c as input to the flow module should indeed allow the extraction of backward flow, and be sufficient for self-supervised learning of the flow module without requiring flow inversion. However, to apply Eq. (1) and to complete the forward pass of the decoder, we still need the forward flow to warp e_c and update d_s, which again requires flow inversion if computed from the backward flow. Another solution would be to go through the flow module twice at train time to compute the flow in both directions, something we did not do as it would slow down training. The chosen solution estimates the forward flow in the flow module and relies on flow inversion to obtain the target flow for self-supervised learning. We will discuss these issues further to better motivate the need for flow inversion.

---

> > ### Comment · Reviewer_Q5Cg · 2021-08-25
> > **Reply to the authors' response**
> >
> > Thanks for preparing the thorough rebuttal.
> >
> > The authors' responses have addressed most of this reviewer's concerns, which increases a rating.
> > The authors are strongly required to reflect all the rebuttal responses in the paper; especially missing reference, acknowledging the conventional works, the additional results, and the preliminary results with the SPADE version.
> > More discussion about the flow module is recommended.
> >
> >
> > <Additional comment>
> >
> > - Flow inversion approximation: The computation time comparison between the forward pass vs. the flow inversion is recommended to be presented. Could the authors answer this question?

---

> > > ### Author Response · Authors · 2021-08-27
> > > **RE: Reply to the authors' response**
> > >
> > > We appreciate the positive feedback on our rebuttal. We will proceed as the reviewer suggests when preparing our revision.
> > >
> > > **Flow inversion approximation** At resolution 256x256, the forward pass of the flow module for 16 images takes around 0.15s on a GPU.
> > > Flow inversion runs on parallelized CPU processes as part of data loading (1 image per process) which takes around 0.12s at the same resolution. In our setup, GPU consumption is the limiting factor for speed. Hence, flow inversion does not slow down training despite the extra CPU cost. In the contrary, a second forward pass in the flow module would inevitably increase the GPU consumption and also the training time.

---

### Author Response · Authors · 2021-08-10
**Overall Response from the Authors to all Reviewers**

We thank the reviewers for their constructive feedback, including positive comments about the novelty of our work, ablation study quality, and state-of-the-art quantitative results. They also express some concerns, which we address in this rebuttal.

Detailed responses have been given to each reviewer. We summarize the major points here:

* All reviewers have expressed the need for an assessment of the potential for diversity in our generated videos. The corresponding results are given in the individual responses.
* The reviewers have also found it difficult to understand the implementation and training process for the flow module of our method. We will strive to clarify these processes if the paper is accepted, and clarify this point in the individual responses.
*Relevant references are missing in the submission. We acknowledge this, thank the reviewers for their suggestions, and will add relevant references.
*The reviewers asked for more qualitative results, together with discussion on improvements over prior works. We provide them here through anonymized links (as explicitly authorized by the NeurIPS guidelines this year) and will include them in our revision.

---

### Decision · Program_Chairs · 2021-09-27

**Decision:**

Accept (Poster)

**Comment:**

The paper proposed a conditional video synthesis model based on the vision transformer architecture and the VQVAE architecture All the reviewers considered the paper above the bar. The rebuttal successfully answered several questions raised by the reviewers, with two reviewers upgraded the score to more positive ratings. Overall, the reviewers consider the paper a welcomed extension of the transformer plus VQVAE paradigm for video synthesis. The quantitative results were convincing. The meta-reviewer agrees with the assessment and would like to recommend its acceptance.